# THE ROLE OF STOCHASTIC ENVIRONMENTS IN ENABLING ADAM

## ABSTRACT

Adaptive optimizers, most notably Adam (and AdamW), are ubiquitous in large-scale first-order training. Yet many theoretical treatments either omit or distort the very features that drive Adam's empirical success, such as momentum and bias correction. Building on recent online-to-nonconvex reductions, we develop a refined discounted-to-nonconvex analysis that respects these features and yields guarantees under a statistically grounded setting. Our key technical contributions are twofold. First, we formalize an online learning with shifting stochastic environments framework that aligns with non-smooth, non-convex stochastic optimization and sharpens how discounted regret translates to optimality conditions. Second, we introduce Adam-FTRL, an online algorithm that exactly matches the plain Adam update in vector form, and prove competitive discounted regret bounds without clipping or unrealistic parameter couplings. Via our conversion, these bounds imply robust convergence of Adam-FTRL to $(\lambda, \rho)$-stationary points, achieving the optimal iteration complexity under favorable environmental stochasticity and shift complexity. The analysis further highlights two environment measures: a normalized signal-to-noise ratio (NSNR) and a discounted shift complexity, which govern convergence behavior and help explain the conditions under which Adam attains its theoretical guarantees.

## 1 INTRODUCTION

Adaptive optimizers, particularly Adam (and AdamW)(Kingma, 2014; Loshchilov, 2017), are essential for the success of large-scale first-order optimization tasks, including the training of large language models (Devlin, 2018; Radford et al., 2019; Bommasani et al., 2021; Touvron et al., 2023; Team et al., 2023). However, despite their widespread practical use, the theoretical foundations underpinning Adam's superior performance remain elusive. Although many efforts have sought to establish convergence rates for Adam comparable to those of stochastic gradient descent (SGD), Adam consistently outperforms SGD empirically. Moreover, current theoretical analyses frequently fall short, neglecting critical components of Adam, such as momentum updates and bias correction terms for the first and second moments. These components, often perceived as technical obstacles, are typically assigned impractical values (e.g., $\beta_1 = \mathcal{O}(1/T)$) or completely omitted from theoretical considerations (Li et al., 2024; Wang et al., 2024). Consequently, a rigorous theoretical framework that fully incorporates these key components and elucidates Adam's empirical advantages remains highly desirable.

Recent advancements in the discounted-to-nonconvex conversion framework provide a promising pathway to better understand Adam's effectiveness. Specifically, Cutkosky et al. (2023) proposed the online-to-nonconvex conversion framework, which establishes a foundational link between non-smooth non-convex optimization and online learning. Building on this, subsequent research(Zhang & Cutkosky, 2024; Ahn & Cutkosky, 2024; Ahn et al., 2024b) introduced the discounted-to-nonconvex conversion framework, offering novel perspectives into the connections between adaptive optimizers and online learning methods. This framework is particularly appealing as it potentially reveals the fundamental mechanisms driving Adam's superior practical performance.

The discounted-to-nonconvex framework comprises two core components: the discounted-to-nonconvex conversion algorithm and an associated online learning algorithm. The theoretical underpinning of the conversion algorithm explicitly relates the optimality conditions of non-smooth non-

convex optimization, such as the gradient norm, to the discounted (static) regret of a corresponding online learner. This reduction bridges optimization guarantees with the extensively studied static regret within the online learning literature, not only clarifying complexity guarantees for stochastic optimization but also providing new aspects of algorithms for stochastic optimization from the classical online learning methods. Critically, this framework formalizes a one-to-one correspondence between online learners and specific optimization algorithms, implying that designing effective non-convex optimizers equates directly to creating online learners that minimize discounted regret.

**Interpretable design, optimality, and adaptivity.** To further support the framework's utility, pioneering work (Ahn et al., 2024b; Ahn & Cutkosky, 2024; Zhang & Cutkosky, 2024) has demonstrated a strong connection between the design principles of certain online learning approaches, such as scale-free Follow-the-Regularized-Leader (FTRL) and $\beta$-FTRL (Orabona & Pál, 2018; Zhang et al., 2024a), and the essential features in popular adaptive optimizers for stochastic optimization. Meanwhile, leveraging the discounted regret bounds and the theoretical guarantees provided by the discounted-to-nonconvex conversion algorithm, optimization guarantees for non-smooth non-convex stochastic optimization problems have been successfully established. For instance, Ahn & Cutkosky (2024) showed that $\beta$-FTRL, which closely resembles Adam, achieves optimal iteration complexity for both non-smooth and smooth non-convex loss functions given appropriate hyperparameter tuning. Furthermore, to justify Adam's performance superiority over SGD, $\beta$-FTRL has been shown to better adapt to problem-dependent properties, i.e., offering stronger theoretical guarantees in scenarios where these problem-dependent properties are unknown (Ahn & Cutkosky, 2024).

In this paper, we advance the discounted-to-nonconvex conversion framework further, achieving notable theoretical improvements. Specifically, our main contributions are:

- *Shifting stochastic environments.* We formalize a segmented i.i.d. model with two interpretable measures: (i) an environmental NSNR controlling signal vs. noise within each segment; and (ii) a discounted shift complexity quantifying how rapidly segments drift under discounting. Both measures surface naturally in the analysis and later in the optimization guarantees.

- *Adam-FTRL (no compromises).* We give an online learner that precisely matches Adam's bias-corrected momentum update. We prove discounted regret bounds that avoid unrealistic operations such as clipping, and keep standard momentum parameters and bias correction terms.

- *Optimal nonconvex iteration complexity.* Via our refined conversion, Adam-FTRL attains optimal iteration complexity for reaching $(\lambda, \rho)$-stationarity under favorable NSNR and shift complexity

## 1.1 RELATED WORK

Significant efforts have been dedicated to understanding Adam's superior performance from two perspectives: *convergence rate* and *adaptivity*: Numerous studies have examined Adam's convergence behavior, demonstrating that it achieves a convergence rate comparable to SGD for convex or smooth nonconvex functions under different stochastic gradient conditions and hyper-parameter configurations (Reddi et al., 2019; Zhou et al., 2018; Alacaoglu et al., 2020; Guo et al., 2021; Zhang et al., 2022; Wang et al., 2024). However, these analyses often fail to capture the contributions of Adam's core components. Moreover, it is well established that under these function assumptions, SGD already achieves the minimax optimal convergence rate (Agarwal et al., 2009; Nesterov et al., 2018; Arjevani et al., 2023). Beyond convergence speed, studying the adaptivity of Adam over complex deep-learning environments is also a popular trend to support the success of Adam. Wang et al. (2023) showed that AdaGrad, a precursor to Adam, can adapt to functions satisfying the generalized smoothness condition (Zhang et al., 2019), while plain SGD may converge arbitrarily slowly. Subsequent work (Li et al., 2024) extended this analysis to Adam, demonstrating its convergence under the generalized smoothness condition. Additionally, Crawshaw et al. (2022) highlighted the theoretical benefits of momentum updates, a component shared by Adam, for SignSGD algorithm under the generalized smoothness condition.

More recently, a novel analysis technique that connects guarantees of non-smooth non-convex optimization with online learning regret has become a trending topic (Cutkosky et al., 2023). Building upon this technique, a series of works build the connections between some variants of well-studied online learning algorithms with adaptive optimization methods (Zhang & Cutkosky, 2024; Ahn

et al., 2024b; Ahn & Cutkosky, 2024), which facilitates understanding the design principles of the core components of adaptive optimization methods.

## 2 PRELIMINARIES

In this section, we firstly introduce the necessary assumptions regarding the function and stochastic gradient for non-smooth non-convex stochastic optimization, adapted from previous studies (Cutkosky et al., 2023; Ahn & Cutkosky, 2024; Zhang & Cutkosky, 2024).

Specifically, we use Assumption 2.1, which has been proven sufficient to develop algorithms that achieve $(\lambda, \rho)$-stationary points. The concept of $(\lambda, \rho)$-stationarity, defined formally in Definition 2.2, is a common notation of optimality for non-smooth non-convex stochastic optimization (Zhang & Cutkosky, 2024; Ahn & Cutkosky, 2024; Zhang et al., 2019; Jordan et al., 2023; Tian et al., 2022). It is worth noting that a $(\lambda, \rho)$-stationary point is a relaxed variant of a Goldstein stationary point (Goldstein, 1977), yet it retains crucial properties that enable conversion to first-order stationary points when the objective function is smooth.

**Assumption 2.1.** Let $F : \mathbb{R}^d \to \mathbb{R}$ be a differentiable function with the following properties:

- The **function** F is bounded below by $\inf_{\mathbf{x}} F(\mathbf{x})$. Meanwhile, defining $\Delta := F(\mathbf{x}_0) - \inf_{\mathbf{x}} F(\mathbf{x})$.

- The **function** F is well-behaved, i.e., $\forall \mathbf{x}$ and $\mathbf{y}$, $F(\mathbf{x}) - F(\mathbf{y}) = \int_0^1 \langle \nabla F(\mathbf{x} + t(\mathbf{y} - \mathbf{x}), \mathbf{y} - \mathbf{x}) \rangle dt$.

- The **function** F is $G$-Lipshitz, i.e., $\forall \mathbf{x}, \|\nabla F(\mathbf{x})\| \leq G$.

- The **stochastic gradient** $\mathbf{g} \leftarrow \text{StoGrad}(\mathbf{x}, r)$ for randomness $r \in \mathcal{Z}$, and $\forall \mathbf{x}$ satisfies $\mathbb{E}[\mathbf{g}] = \nabla F(\mathbf{x})$ and $\mathbb{E}[\|\mathbf{g} - \nabla F(\mathbf{x})\|^2] \leq \sigma^2$. Note a quick result $\mathbb{E}[\|\mathbf{g}\|^2] \leq G^2 + \sigma^2$.

**Definition 2.2** ($\lambda, \rho$-stationary point). Supposing $F(\cdot) : \mathbb{R}^d \to \mathbb{R}$ is differentiable. Then $\mathbf{x}$ is a $(\lambda, \rho)$-stationary point of $F$ if $\|\nabla F(\mathbf{x})\|^{[\lambda]} \leq \rho$ where $\|\nabla F(\mathbf{x})\|^{[\lambda]} := \inf_{p \in \mathcal{P}(\mathbb{R}^d), \mathbb{E}_{\mathbf{y} \sim p}[\mathbf{y}] = \mathbf{x}} \{\|\mathbb{E}[\nabla F(\mathbf{y})]\| + \lambda \mathbb{E}[\|\mathbf{y} - \mathbf{x}\|^2]\}$.

We now briefly introduce fundamental concepts of online learning and essential definitions of regret, which underpin our analysis. Online Linear Optimization (OLO) is an iterative algorithm: at $s$-th step of $t$ total rounds, the online learner selects an action $\mathbf{z}_s \in \mathbb{R}^d$ and then receives a linear loss $\ell_s(\cdot) := \langle \mathbf{v}_s, \cdot \rangle$, where $\mathbf{v}_s \in \mathbb{R}^d$ is a cost vector revealed by the environment. The learner aims to minimize static regret, defined as the cumulative difference between the learner's losses and those of an arbitrary comparator $\mathbf{u}_t \in \mathcal{D} \subseteq \mathbb{R}^d$:

$$\text{Regret}_t(\mathbf{u}_t) := \sum_{s=1}^{t} (\ell_s(\mathbf{z}_s) - \ell_s(\mathbf{u}_t)) = \sum_{s=1}^{t} \langle \mathbf{v}_s, \mathbf{z}_s - \mathbf{u}_t \rangle.$$

In this work, we specifically focus on a bounded domain scenario, where $\mathcal{D}$ is the $d$-dimensional $L_2$-ball with radius $D$.

**Shifting comparators.** Additionally, we address the online linear learning setting with shifting comparators. This more challenging setting allows for a sequence of arbitrary comparators $\{\mathbf{u}_k\}_{k \in [K]}$, and the regret in this scenario is defined as:

$$\text{Regret}_t(\mathbf{u}_1, \cdots, \mathbf{u}_K) := \sum_{k=1}^{K} \sum_{s=(k-1)t+1}^{kt} \langle \mathbf{v}_s, \mathbf{z}_s - \mathbf{u}_k \rangle.$$

The concept of shifting comparators effectively models situations where the environment that generates the cost vectors evolves over time, and different comparators are best for different subsequences of cost vectors or fit each segmented environment best (Herbster & Warmuth, 1998).

**Discounting static regret.** Finally, we introduce the concept of $\beta$-discounted static regret, essential to our framework. Given an algorithm that discounts the loss by $\beta^{t-s}$ at $s$-th step over $t$ rounds, i.e., $\ell_t^{[\beta]}(\cdot) = \beta^{t-s} \ell_t(\cdot)$, the associated $\beta$-discounted regret is defined as:

$$\text{Regret}_t^{[\beta]}(\mathbf{u}_t) := \sum_{s=1}^{t} \langle \beta^{t-s} \mathbf{v}_s, \mathbf{z}_s - \mathbf{u}_t \rangle.$$

For further details about online learning, readers may refer to Orabona (2019); Cutkosky et al. (2023).

---

**Algorithm 1** Discounted-to-nonconvex conversion algorithm (Zhang & Cutkosky, 2024)

---

1: **Input:** Initial point $\mathbf{x}_0$, $T$, online learner $\mathcal{A}$ outputting $\mathbf{z}$, and discounting factor $\beta \in (0, 1)$
2: **for** $t = 1$ **to** $T$ **do**
3:     Receive $\mathbf{z}_t$ from $\mathcal{A}$
4:     Update $\mathbf{x}_t \leftarrow \mathbf{x}_{t-1} + \rho_t \mathbf{z}_t$, where $\rho_t \sim \text{Exp}(1)$ i.i.d.
5:     Compute $\mathbf{g}_t \leftarrow \text{StoGrad}(\mathbf{x}_t, \mathbf{r}_t)$ with freshly sampled randomness $\mathbf{r}_t$
6:     Send $\ell_t^{[\beta]}(\mathbf{z}_t) := \langle \beta^{T-t} \mathbf{g}_t, \mathbf{z}_t \rangle$ to $\mathcal{A}$
7:     $\widetilde{\mathbf{x}}_t \leftarrow \frac{\beta - \beta^t}{1 - \beta^t} \widetilde{\mathbf{x}}_{t-1} + \frac{1-\beta}{1-\beta^t} \mathbf{x}_t;$. {For output.}
8: **end for**
9: Return $\mathbf{x}_{\text{output}}$ where $\mathbf{x}_{\text{output}}$ is distributed over $\{\widetilde{\mathbf{x}}_t\}_{t \in [T]}$.

---

## 3 BACKGROUND

### 3.1 DISCOUNTED-TO-NONCONVEX CONVERSION

This work builds upon the discounted-to-nonconvex conversion recently developed by Zhang & Cutkosky (2024); Ahn & Cutkosky (2024), outlined in Algorithm 1. A central lemma (Lemma 7 Ahn & Cutkosky (2024)) shows that the averaged expected gradient norm can be controlled by a time-average of discounted regrets: averaged-expected gradient norm $\lesssim \frac{1}{T} \sum_{t=1}^T \mathbb{E} \left[ \text{Regret}_t^{[\beta]}(\mathbf{u}_t) \right]$

Intuitively, this conversion views the non-smooth non-convex optimization process over a total of $T$ rounds as online learning with shifting comparators. Each segment within the total $T$ rounds corresponds to an environment with its best-fitting comparator. Therefore, the classical static regret framework from online learning naturally extends to the classical optimization guarantee, facilitating the analysis of the entire optimization process. More detailed discussions of this conversion can be found in Section 3 of Zhang & Cutkosky (2024) and Section 3 of Cutkosky et al. (2023).

### 3.2 ONLINE LEARNING ALGORITHM: SCALE-FREE FTRL AND $\beta$-FTRL

The scale-free Follow-the-Regularized-Leader (FTRL) algorithm is a well-established algorithm in online learning (Orabona & Pál, 2018). In terms of the increment selecting strategy, scale-free FTRL tracks the history of linear loss function and adjusts its prediction based on past cost vector sequence $\{\mathbf{v}_s\}_{s \in [t-1]}$ and on the selected regularizer $\{\frac{1}{2\alpha_t} || \cdot ||^2\}$ at $t$-th step, effectively leveraging all past information to refine future predictions. The scale-free FTRL is presented in Algorithm 2 (scale-free FTRL). Ahn et al. (2024b); Ahn & Cutkosky (2024) provide a key insight: incorporating $\beta$-discounted regret, which better describes the learning goal of online learners in dynamic environments (Zhang et al., 2024b; Jacobsen & Cutkosky, 2024), into the design of scale-free FTRL algorithm leads to a formulation nearly resembling the Adam optimizer. By substituting $\mathbf{v}_s$ in scale-free FTRL with $\beta^{-s} \mathbf{v}_s$, the result is the $\beta$-FTRL, presented in Algorithm 2 ($\beta$-FTRL).

### 3.3 LIMITATIONS OF PRIOR FTRL VARIANTS

Despite their theoretical strengths, existing FTRL variants have notable practical limitations. Specifically, the derivation of discounted regret bound for $\beta$-FTRL heavily depends on the clipping operation $\text{clip}_D(\cdot)$ applied to incremental updates, enforces restrictive hyperparameter settings such as $\beta_2 = \beta^2$ and $\beta_1 = \beta$, and omits critical components of Adam such as bias-correction terms.

The clipping operation $\text{clip}_D(\mathbf{x}) := \mathbf{x} \min(D/||\mathbf{x}||, 1)$ is neither practically realistic (due to the unknown problem-dependent property $D$) nor used in standard Adam implementations. Similarly, setting $\beta_2 = \beta_1^2$ or eliminating the bias-correction terms simplifies theoretical analysis but diverges significantly from empirical practice and can even degrade performance in realistic deployment scenarios. These algorithmic compromises further introduce substantial theoretical challenges; notably, a hard threshold operation basically converts the standard bounded-variance assumptions into stronger, bounded-supremum variance assumptions in a theoretical sense.

Rather than the algorithmic compromises, our approach keeps Adam intact and instead analyzes a more realistic environment, potentially revealing how the environment enables Adam theoretically.

---

**Algorithm 2** Scale-free FTRL, $\beta$-FTRL, proposed Adam-FTRL

---

1: **Input:** Regularizer $\left\{\frac{1}{2\alpha_t}||\cdot||^2\right\} : \mathbb{R}^d \to \mathbb{R}$, the bounded domain $\mathcal{D}$, and $\beta \equiv \beta_1, \beta_2 \in (0, 1)$.

2: **for** $t = 1$ **to** $T$ **do**

3:  •  scale-free: $\mathbf{z}_t = \underset{\mathbf{z} \in \mathcal{D}}{\arg\min} \left[ \frac{||\mathbf{z}||^2}{2\alpha_t} + \sum_{s=1}^{t-1} \langle \mathbf{v}_s, \mathbf{z} \rangle \right] = -\text{clip}_D \left( \eta \frac{\sum_{s=1}^{t-1} \mathbf{v}_s}{\sqrt{\sum_{s=1}^{t-1} ||\mathbf{v}_s||^2}} \right)$[a]

   •  $\beta$-FTRL: $\mathbf{z}_t = \underset{\mathbf{z} \in \mathcal{D}}{\arg\min} \left[ \frac{||\mathbf{z}||^2}{2\alpha_t} + \sum_{s=1}^{t-1} \langle \beta^{-s} \mathbf{v}_s, \mathbf{z} \rangle \right] = -\text{clip}_D \left( \eta \frac{\sum_{s=1}^{t-1} \beta^{-s} \mathbf{v}_s}{\sqrt{\sum_{s=1}^{t-1} ||\beta^{-s} \mathbf{v}_s||^2}} \right)$[b]

   •  Adam-FTRL: $\mathbf{z}_t = \arg\min \left[ \frac{||\mathbf{z}||^2}{2\alpha_t} + (1 - \beta_1) \sum_{s=1}^{t-1} \langle \beta_1^{t-1-s} \mathbf{v}_s, \mathbf{z} \rangle \right]$

   $= -\frac{\eta \sqrt{1 - \beta_2^{t-1}}}{1 - \beta_1^{t-1}} \frac{(1 - \beta_1) \sum_{s=1}^{t-1} \beta_1^{t-1-s} \mathbf{v}_s}{\sqrt{(1 - \beta_2) \sum_{s=1}^{t-1} \beta_2^{t-1-s} ||\mathbf{v}_s||^2}}$ [c]

4:   Receive $\ell_t(\cdot) = \langle \mathbf{v}_t, \cdot \rangle$

5: **end for**

---

[a] By selecting $\alpha_t$ as $\frac{\eta}{\sqrt{\sum_{s=1}^{t-1} ||\mathbf{v}_s||^2}}$. And $\text{clip}_D(\mathbf{x}) := \mathbf{x} \min(D/||\mathbf{x}||, 1)$.

[b] By selecting $\alpha_t$ as $\frac{\eta}{\sqrt{\sum_{s=1}^{t-1} ||\beta^{-s} \mathbf{v}_s||^2}}$.  [c] By selecting $\alpha_t$ as $\frac{\eta \sqrt{1 - \beta_2^{t-1}}}{1 - \beta_1^{t-1}} \frac{1}{\sqrt{(1 - \beta_2) \sum_{s=1}^{t-1} \beta_2^{t-1-s} ||\mathbf{v}_s||^2}}$.

---

### 3.4 PAPER ORGANIZATION

Section 4 defines shifting stochastic environments and their measures. Section 5 analyzes Adam-FTRL's discounted regret in a static segment. Section 6 converts regret to nonconvex guarantees and interprets the results.

## 4 ONLINE LEARNING WITH SHIFTING STOCHASTIC ENVIRONMENTS

The discounted-to-nonconvex conversion framework, despite its potential in analyzing algorithms for non-smooth non-convex optimization, is currently limited due to its reliance on a purely adversarial online learning setting. In this section, we introduce a *shifting stochastic environment* for online learning, transitioning from arbitrary sequences of cost vectors to more structured sequences. Subsequently, we discuss the conversion from online learning with shifting stochastic environments to non-smooth non-convex stochastic optimization, highlighting the role of segmented environments.

### 4.1 SHIFTING STOCHASTIC ENVIRONMENT: DEFINITION AND ASSUMPTIONS

In the classical adversarial online learning framework, the environment selects cost vectors arbitrarily. However, there are variations of online learning (sometimes called stochastic online learning or statistical online learning), which are studied under a stochastic (i.i.d.) environment rather than in a purely adversarial setting. This view of online learning potentially connects online learning methods to stochastic gradient methods and facilitates translating regret bounds into guarantees in stochastic optimization. We first define the shifting stochastic environment:

**Definition 4.1** (Shifting stochastic environment). Consider a sequence of cost vectors $\{\mathbf{v}_t\}_{t \in [T]}$, each drawn independently from a sequence of distributions with mean vectors $\{\boldsymbol{\mu}_t\}_{t \in [T]}$ and finite covariance matrices $\{\boldsymbol{\Sigma}_t\}_{t \in [T]}$, where $\boldsymbol{\mu}_t \in \mathbb{R}^d$ and $\boldsymbol{\Sigma}_t \succ 0, \forall t$. Thus, we immediately have $\mathbb{E}\left[||\mathbf{v}_t - \boldsymbol{\mu}_t||^2\right] = \text{Tr}(\boldsymbol{\Sigma}_t)$ and $\mathbb{E}\left[||\mathbf{v}_t||^2\right] = \text{Tr}(\boldsymbol{\Sigma}_t) + ||\boldsymbol{\mu}_t||^2$.

Given our focus on studying the stochastic optimization method Adam through the online learning techniques, regarding the online learning environment, we make corresponding assumptions that are not stronger than the common assumptions in stochastic optimization.

Assumption 4.2 implies the standard assumptions in stochastic optimization, i.e., unbiased stochastic gradient and bounded variance.

**Assumption 4.2** (Standard assumption). Using notations from Definition 4.1. $\forall t$, $||\boldsymbol{\mu}_t|| \leq G$, $\mathbb{E}\left[||\mathbf{v}_t - \boldsymbol{\mu}_t||^2\right] \leq \sigma^2$. Consequently $\mathbb{E}\left[||\mathbf{v}_t||^2\right] \leq (\sigma + G)^2$.

Beyond the standard definitions and assumptions, we further introduce a derived measure from Definition 4.1, the environmental Normalized Signal-to-Noise Ratio (NSNR), to more effectively characterize the stochastic environments considered in this work:

**Definition 4.3** (Environmental NSNR)**.** Using notations from Definition 4.1. we define a derived constant $c_1 \geq 0$ such that $\forall t$, $\frac{\|\boldsymbol{\mu}_t\|^2}{\|\boldsymbol{\mu}_t\|^2 + \mathrm{Tr}(\boldsymbol{\Sigma}_t)} \leq c_1(1 - \beta_1)$ where $\beta_1 \in (0, 1)$. Thus, $c_1(1 - \beta_1)$ serves as an upper bound for the NSNR in each segmented stochastic environment.

Clearly, such a constant $c_1$ always exists (e.g., $c_1 = \frac{1}{1-\beta_1}$ in the worst case). With such a definition, we aim to emphasize the impacts of $c_1$ and introduce it into the final convergence statement, further discussed in Section 6.2.

### 4.1.1 SLIDING STATIC SEGMENTS AND ENVIRONMENTAL SHIFTING COMPLEXITY

From arbitrary cost vectors to the more structured forms described above, the environment does not necessarily become easier for online learning algorithms. In particular, when minimizing regret, algorithms still face similar technical challenges due to the drifting mean of the hidden distributions underlying the stochastic environment. However, this reformulation allows us to develop strategies that alleviate the burden of directly handling the drifting mean within online learning algorithms. Specifically, we introduce a sliding static segments perspective on shifting stochastic environments, together with a corresponding notion of restricted shifting complexity.

Formally, we model a $T$-round shifting stochastic environment as a sequence of static segments: partition $[T]$ into $N$ segments with endpoints $1 = t_0 < t_1 < \cdots < t_N = T$.[1] Within each segment, the hidden distribution is fixed, and the environment is therefore static.

Viewing the environment as sliding static segments naturally highlights the necessity of restricting the shifting complexity. Without such restrictions, for instance, if $N \equiv T$, the formulation collapses into triviality. Beyond this, two further considerations arise: (i) in a fully adversarial environment, historical information about cost vectors provides no predictive power, reducing algorithm design to triviality; and (ii) such adversarial environments may preclude any convergence guarantees.

Motivated by the standard complexity measure for shifting comparators, the path length $P := \sum_{t=1}^{T-1} \|\mathbf{u}_t - \mathbf{u}_{t-1}\|$, which quantifies the complexity of a comparator sequence $\mathbf{u}_{0:T-1}$ (Herbster & Warmuth, 2001), we propose a discounted variant that captures the shifting complexity of environments.

**Definition 4.4** (Environmental shifting complexity)**.** Given a set of segment indices $\{t_i\}_{i \in [N-1]}$, define $c_T = \beta_1^{T-t_1+1}\sqrt{1 - \beta_1^{t_1-1}} + \beta_1^{T-t_2+1}\sqrt{1 - \beta_1^{t_2-t_1}} + \cdots + \sqrt{1 - \beta_1^{T-t_{N-1}}}$. It follows that $1 - \beta_1^T \leq c_T \leq \frac{1-\beta_1^T}{\sqrt{1-\beta_1}}$ where $\beta_1 \in (0, 1)$. We then define a constant $c_0$ such that the segmentation $\{t_i\}_{i \in [N-1]}$ ensures $c_T \leq c_0$.

Importantly, $c_0$ is not a trivial constant that can grow arbitrarily large. In fact, under any shifting complexity, $c_T$ can be bounded by such as $c_0 = \frac{1}{\sqrt{1-\beta_1}}$. This definition emphasizes the role of $c_0$ in characterizing the environment, and we will incorporate it into the final convergence statement in Section 6.2.

Moreover, both $c_0$ and another constant $c_1$ are intrinsically tied to the parameter $\beta_1$. These constants play pivotal roles in the ultimate convergence result, where further assumptions allow us to explicitly explore the role of stochastic environments in enabling Adam, consistent with the central theme of this work.

### 4.2 FROM SHIFTING STOCHASTIC ENVIRONMENT TO NON-CONVEX CONVERSION

Building on the above environment modeling, we establish that the averaged expected gradient norm can be upper-bounded by the discounted regret of online learning algorithms under a static stochastic environment. This result is formally stated in Theorem 4.6 with its supporting result Lemma 4.5.

---

[1]The starting and ending indices of each segment are: segment 1, $[1, \ldots, t_1-1]$; segment 2, $[t_1, \ldots, t_2-1]$; $\ldots$; segment $N$, $[t_{N-1}, T]$. For notational consistency, we set $t_0 = 1$ and $t_N = T$.

**Lemma 4.5.** *Consider the iterations generated as in Algorithm 1, $\forall t \in [T]$, we split $t$-shifting stochastic environments into $n$-segments, it is guaranteed that*

$$\mathbb{E}\left[\sum_{s=1}^{t}\beta_1^{t-s}\left(F(\mathbf{x}_s)-F(\mathbf{x}_{s-1})\right)\right] \leq -\frac{(1-\beta_1^t)D}{1-\beta_1}\inf_{s\in[t]}\|\nabla F(\mathbf{x}_s)\| + c_0\frac{\sigma D}{\sqrt{1-\beta_1}} + c_0\sup_{i\in[n]}\frac{\mathbb{E}\left[Regret_{t_i}^{[\beta]}(\mathbf{u}_{t_i})\right]}{\sqrt{1-\beta_1^{t_i-t_{i-1}}}},$$

*where $Regret_{t_i}^{[\beta]}(\mathbf{u}_{t_{i-1}}) = \sum_{s=t_{i-1}}^{t_i-1}\langle\beta_1^{t_i-1-s}\mathbf{v}_s, \mathbf{z}_s - \mathbf{u}_{t_i}\rangle$, where $\mathbf{v}_{t_{i-1}}, \cdots, \mathbf{v}_{t_i-1}$ are sampled from $i$-th static stochastic environment.*

**Theorem 4.6** (Static stochastic environment discounted-to-nonconvex conversion). *Supposing that $F$ satisfies Assumption 2.1, the Algorithm 1 guarantees*

$$\mathbb{E}_\tau\|\mathbb{E}_{\widetilde{\mathbf{X}}_\tau}\nabla F(\widetilde{\mathbf{X}}_\tau)\| \leq \frac{c_0}{DT}\left(\beta_1\mathbb{E}\left[\sup_{i\in[N]}\frac{\mathbb{E}\left[Regret_{t_i}^{[\beta]}(\mathbf{u}_{t_i})\right]}{\sqrt{1-\beta_1^{t_i-t_{i-1}}}}\right] + (1-\beta_1)\sum_{t=1}^{T}\mathbb{E}\left[\sup_{i\in[n]}\frac{\mathbb{E}\left[Regret_{t_i}^{[\beta]}(\mathbf{u}_{t_i})\right]}{\sqrt{1-\beta_1^{t_i-t_{i-1}}}}\right]\right)$$
$$+\frac{\Delta}{DT}+\frac{c_0\sigma}{T\sqrt{1-\beta_1}}+c_0\sigma\sqrt{1-\beta_1}.$$

*where $\widetilde{\mathbf{X}}_\tau$ is distributed over $\{\widetilde{\mathbf{x}}_t\}_{t=1}^{T}$ as $\mathbb{P}(\tau=t) = \begin{cases} \frac{1-\beta_1^t}{T} & \text{for } t=1,\cdots,T-1, \\ \frac{1}{1-\beta_1}\frac{1-\beta_1^T}{T} & \text{for } t=T. \end{cases}$*

The proofs of Lemma 4.5 and Theorem 4.6 is provided in Appendix A.1. Theorem 4.6 shows that the central quantity in the optimality condition (Definition 2.2) is controlled by discounted regret under a static stochastic environment, along with several additional terms. Consequently, the theorem enables translating online learning guarantees into non-convex optimization guarantees, in the spirit of existing online-to-nonconvex conversion results (Cutkosky et al., 2023; Zhang & Cutkosky, 2024). Notably, Theorem 4.6 achieves this by explicitly trading off a tight regret bound for a more favorable setting in which online learning algorithms operate.

*Remark* 4.7. Our result, which considers a supremum-type discounted regret $\sup_{i\in[n]}\frac{\mathbb{E}\left[Regret_{t_i}^{[\beta]}(\mathbf{u}_{t_i})\right]}{\sqrt{1-\beta_1^{t_i-t_{i-1}}}}$ where the associated cost vectors are sampled from a static stochastic environment, contrasts with existing results that bound the discounted regret $\mathbb{E}\left[Regret^{[\beta]}(\mathbf{u}_t)\right]$ for arbitrary cost vector sequences (Ahn & Cutkosky, 2024). The difference lies in an additional coefficient, $\frac{1}{\sqrt{1-\beta_1^{\text{iters}}}}$, where *iters* denotes the number of iterations within a given environmental segment. While this factor requires a tighter bound on discounted regret, the trade-off is advantageous: it enables analyzing the algorithm in a simplified, structured environment.

With these foundations established, we next introduce our proposed algorithm, rigorously analyze its performance under a static stochastic environment, and present comprehensive theoretical guarantees.

# 5 ONLINE LEARNING METHOD: ADAM-FTRL UNDER STATIC STOCHASTIC ENVIRONMENT

In this section, we first describe the algorithmic structure of the proposed Adam-FTRL approach. We then establish theoretical guarantees, focusing explicitly on its performance in terms of $\beta$-discounted regret under a specific static stochastic environment.

## 5.1 ALGORITHM: ADAM-FTRL

We formally introduce Adam-FTRL in Algorithm 2. Its update rule for the incremental variable $\mathbf{z}_t$ is given by

$$\mathbf{z}_t = -\eta\frac{\sqrt{1-\beta_2^{t-1}}}{1-\beta_1^{t-1}}\frac{(1-\beta_1)\sum_{s=1}^{t-1}\beta_1^{t-1-s}\mathbf{v}_s}{\sqrt{(1-\beta_2)\sum_{s=1}^{t-1}\beta_2^{t-1-s}\|\mathbf{v}_s\|^2}} \tag{1}$$

This update rule matches that of the well-known Adam algorithm (excluding the numerical stability parameter $\epsilon$). To ensure proper initialization and avoid degenerate cases, we explicitly define the algorithm's behavior when the denominator in equation 1 becomes zero: in such cases, the update is set to zero. For example, $\mathbf{z}_1 = 0$, and equation 1 takes effect from $t = 2$ onward.

## 5.2 INCREMENTAL AND DISCOUNTED REGRET OF ADAM-FTRL

We begin by characterizing the incremental $\mathbf{z}_t$, summarized in the following lemma.

**Lemma 5.1.** *When* $\beta_1 \le \beta_2 \le \sqrt{\beta_1}$, *Adam-FTRL satisfies* $\|\mathbf{z}_t\|^2 \le \frac{4\eta^2}{1-\beta_1}$.

The proof of Lemma 5.1 is provided in Appendix A.2, and may be of independent interest.

Next, we establish a rigorous bound for the $\beta$-discounted regret of Adam-FTRL, improving upon prior results by (i) eliminating the need for unrealistic clipping operations, and (ii) achieving a closer theoretical alignment with plain Adam.

**Theorem 5.2** (Discounted regret of Adam-FTRL). *Assume Assumption 4.2 holds. For a $\beta$-discounted loss sequence $\beta_1^{t-1}\mathbf{v}_1, \cdots, \beta_1^{t-t}\mathbf{v}_t$, comparator $\mathbf{u}_t \in \mathcal{D}$, i.e., $\|\mathbf{u}_t\| \le D$, Adam-FTRL guarantees* $\mathbb{E}\left[Regret_t^{[\beta]}(\mathbf{u}_t)\right] = 3\sqrt{1+2c_1}D\sqrt{\frac{1-\beta_1^t}{1-\beta_1}}\sqrt{Tr(\mathbf{\Sigma}_t) + \|\boldsymbol{\mu}_t\|^2}$ *when the learning rate is chosen as* $\eta = \sqrt{1-\beta_1}D$.

The proof of Theorem 5.2 is presented in Appendix A.2.1.

An immediate consequence of Lemma 5.1 and Theorem 5.2 is the following boundedness property of the incremental: $\|\mathbf{z}_t\|^2 \le 4D^2$.

*Remark* 5.3. Lemma 5.1, together with the optimal learning rate in Theorem 5.2, establishes the boundedness of the incremental updates. This boundedness is crucial for the subsequent theoretical analysis. Moreover, it shows that Adam-FTRL achieves the same theoretical effect as the previously used (but practically unrealistic) clipping operation $\text{Clip}_D(\cdot)$. In this way, our analysis aligns theoretical guarantees more closely with practical implementations of Adam.

# 6 OPTIMIZATION GUARANTEE OF ADAM-FTRL

We now establish the optimization guarantee of Adam-FTRL for non-smooth non-convex optimization, providing a formal analysis of its convergence behavior.

## 6.1 CONVERGENCE BEHAVIOR OF ADAM-FTRL

Building on the conversion guarantee (Theorem 4.6) and the discounted regret bound (Theorem 5.2), we present the non-convex optimization guarantee of Adam-FTRL in terms of $(\lambda, \rho)$-stationarity.

**Theorem 6.1.** *Assume that $F$ satisfies Assumption 2.1 and consider any $\lambda > 0$. Algorithm 1, when selecting $\mathcal{A}$ as Adam-FTRL, guarantees*

$$\mathbb{E}_\tau\left[\|\nabla F(\widetilde{\mathbf{x}})\|^{[\lambda]}\right] \le (2 + 4c_0\sqrt{1+2c_1})\epsilon$$

*when the parameters are chosen as* $T = \max\left(8\Delta\lambda^{\frac{1}{2}}(G+\sigma)^2\epsilon^{-\frac{7}{2}}, 4c_0\sqrt{1+2c_1}(G+\sigma)^2\epsilon^{-2}\right)$, $\beta_1 = 1 - \frac{\epsilon^2}{(G+\sigma)^2}$, *and* $D = \frac{1}{8}(1-\beta_1)\lambda^{-\frac{1}{2}}\epsilon^{\frac{1}{2}}$. *I.e.,* $\widetilde{\mathbf{x}}$ *is a* $(\lambda, (2+4c_0\sqrt{1+2c_1})\epsilon)$-*stationary point.*

The proof of Theorem 6.1 is provided in Appendix A.3.

## 6.2 INTERPRETATION AND PRACTICAL IMPLICATIONS

We now interpret the convergence guarantee of Adam-FTRL. For clarity, we first treat the constants $c_0$ and $c_1$ as values independent of $\beta_1$. As shown in Zhang & Cutkosky (2024), the lower bound on iteration complexity for finding $(\lambda, \rho)$-stationary points is $\Omega\left(\Delta\lambda^{1/2}(G+\sigma)^2\epsilon^{-7/2}\right)$. In Theorem 6.1, the dominant $\epsilon$-dependence yields $T = \mathcal{O}(\Delta\lambda^{\frac{1}{2}}(G+\sigma)^2\epsilon^{-\frac{7}{2}})$ up to rescaling, thereby achieving the optimal iteration complexity.

The convergence guarantees hinge on two key environmental constants: (i) the constant $c_1$, associated with the normalized noise-to-signal ratio (NSNR), and (ii) the constant $c_0$, which captures the shifting complexity of the environment.

- **Environmental NSNR.** The results show that a sufficient level of stochasticity is required to regulate the NSNR near $1 - \beta_1$, which in turn enables the algorithm to achieve the optimal convergence rate. In practice, tuning the batch size effectively controls the level of stochasticity. Our theory suggests that Adam does not always benefit from larger batch sizes: empirical evidence indicates that Adam without added gradient noise may underperform both in training and test performance (Neelakantan et al., 2015; Staib et al., 2019). While the precise trade-off depends on problem-specific characteristics, our framework provides theoretical support for this phenomenon.

  In the worst case, when $c_1 = \frac{1}{1-\beta_1} \to \frac{(G+\sigma)^2}{\epsilon^2}$, the guarantee degrades to $\mathbb{E}_\tau \left[ \|\nabla F(\widetilde{\mathbf{x}})\|^{[\lambda]} \right] = \mathcal{O}(G + \sigma)$, meaning that the algorithm converges only to a neighborhood of a stationary point, with error characterized by $G + \sigma$. We attribute this residual error to the fixed bias inherent in Adam-type updates in an exact gradient descent environment, in the absence of further assumptions.

- **Environmental shifting complexity.** Recall Definition 4.4. When the shifting complexity $c_T$ remains small, the constant $c_0$ is also small, ensuring the optimal convergence rate.

  In the worst case, when $c_0 = \frac{1}{\sqrt{1-\beta_1}} \to \frac{G+\sigma}{\epsilon}$, we again obtain $\mathbb{E}_\tau \left[ \|\nabla F(\widetilde{\mathbf{x}})\|^{[\lambda]} \right] = \mathcal{O}(G + \sigma)$ mirroring the worst-case result for the environmental NSNR. Intuitively, this trade-off is unavoidable: if the historical gradient sequence carries no predictive power for future gradients, meaningful convergence cannot be expected.

# 7 DISCUSSION

Our analysis of Adam builds upon the discounted-to-nonconvex conversion framework, but refines it by introducing the perspective of online learning in shifting stochastic environments. As shown in Section 5, this refinement eliminates the non-trivial algorithmic compromises present in existing FTRL variants and demonstrates that plain Adam can, under favorable environmental conditions, achieve the optimal iteration complexity. Moreover, our study identifies two key environmental factors: normalized noise-to-signal ratio (NSNR) and shifting complexity that fundamentally shape convergence behavior.

Despite these contributions, several limitations and open questions remain:

- **Constraining the adversary.** Like path length, which limits the cumulative movement of a comparator sequence (Herbster & Warmuth, 2001), our sliding static segments view provides a way to restrict the power of the environment. However, it makes a stronger assumption than path-length or variation-budget settings by requiring well-defined static segments, an active area of research (Wu et al., 2023). This abstraction may not reflect scenarios where drift is gradual, motivating extensions to more general smooth-drift environments.

- **Dependence on $c_0$ and $c_1$.** The environmental constants $c_0$ and $c_1$ provide valuable theoretical insight into the role of stochasticity and shifting complexity. However, they currently lack practical methods for estimation or control. Bridging this gap, by connecting them to observable quantities such as batch size, variance, or empirical drift, represents an important direction for future work.

- A further distinction arises between the coordinate-wise updates used in practical Adam and the vector updates analyzed in our framework. Bridging this gap analytically typically requires coordinate-wise assumptions on stochastic gradients and objective functions (Crawshaw et al., 2022; Zhuang et al., 2022), such as coordinate-wise $G$-Lipschitz conditions. Under such assumptions, our analysis and results extend naturally to coordinate-wise settings, though we leave this extension to future work.

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

## A  APPENDIX

### LLM USAGE DISCLOSURE

A large language model (ChatGPT, GPT-5, OpenAI) was used solely to polish the writing of this manuscript, including grammar correction and stylistic refinement.

### A.1  PROOFS FOR SECTION 4

**Lemma 4.5.** *Consider the iterations generated as in Algorithm 1, $\forall t \in [T]$, we split $t$-shifting stochastic environments into $n$-segments, it is guaranteed that*

$$\mathbb{E}\left[\sum_{s=1}^{t} \beta_1^{t-s}\left(F(\mathbf{x}_s) - F(\mathbf{x}_{s-1})\right)\right] \le -\frac{(1-\beta_1^t)D}{1-\beta_1} \inf_{s\in[t]} \|\nabla F(\mathbf{x}_s)\| + c_0 \frac{\sigma D}{\sqrt{1-\beta_1}} + c_0 \sup_{i\in[n]} \frac{\mathbb{E}\left[Regret_{t_i}^{[\beta]}(\mathbf{u}_{t_i})\right]}{\sqrt{1-\beta_1^{t_i-t_{i-1}}}},$$

*where $Regret_{t_i}^{[\beta]}(\mathbf{u}_{t_{i-1}}) = \sum_{s=t_{i-1}}^{t_i-1} \langle \beta_1^{t_i-1-s}\mathbf{v}_s, \mathbf{z}_s - \mathbf{u}_{t_i}\rangle$, where $\mathbf{v}_{t_{i-1}}, \cdots, \mathbf{v}_{t_i-1}$ are sampled from $i$-th static stochastic environment.*

*Proof.* We start from

$$\mathbb{E}\left[\sum_{s=1}^{t} \beta^{t-s}\left(F(\mathbf{x}_s) - F(\mathbf{x}_{s-1})\right)\right] \overset{(I)}{=} \mathbb{E}\left[\sum_{s=1}^{t} \beta^{t-s}\langle \nabla F(\mathbf{x}_s), \mathbf{z}_s\rangle\right]$$

$$= \mathbb{E}\left[\sum_{s=1}^{t} \beta^{t-s}\left(\langle \nabla F(\mathbf{x}_s), \mathbf{u}_s\rangle + \langle \nabla F(\mathbf{x}_s) - \mathbf{g}_s, \mathbf{z}_s - \mathbf{u}_s\rangle + \langle \mathbf{g}_s, \mathbf{z}_s - \mathbf{u}_s\rangle\right)\right]$$

$$\overset{(II)}{=} \mathbb{E}\left[\sum_{s=1}^{t} \beta^{t-s}\left(\underbrace{\langle \nabla F(\mathbf{x}_s), \mathbf{u}_s\rangle}_{\text{Part 1}} + \underbrace{\langle \nabla F(\mathbf{x}_s) - \mathbf{g}_s, -\mathbf{u}_s\rangle}_{\text{Part 2}} + \underbrace{\langle \mathbf{g}_s, \mathbf{z}_s - \mathbf{u}_s\rangle}_{\text{Part 3}}\right)\right],$$

where $(I)$ applies Lemma 3.1 in Zhang & Cutkosky (2024); $(II)$ is by the fact $\mathbb{E}\left[\langle \nabla F(\mathbf{x}_s) - \mathbf{g}_s, \mathbf{z}_s\rangle\right] = \mathbb{E}\left[\langle \nabla F(\mathbf{x}_s) - \mathbb{E}\left[\mathbf{g}_s\right], \mathbf{z}_s\rangle\right] = 0$ where where the inner expectation is w.r.t. the randomness in the stochastic gradient oracle and the outer is w.r.t. all other quantities.

Part 1 can be further re-formulated as

$$\mathbb{E}\left[\sum_{s=1}^{t} \beta^{t-s}\langle \nabla F(\mathbf{x}_s), \mathbf{u}_s\rangle\right]$$

$$\overset{(I)}{=} \mathbb{E}\left[\sum_{s=1}^{t_1-1} \beta^{t-s}\langle \nabla F(\mathbf{x}_s), \mathbf{u}_s\rangle + \sum_{s=t_1}^{t_2-1} \beta^{t-s}\langle \nabla F(\mathbf{x}_s), \mathbf{u}_s\rangle + \cdots + \sum_{s=t_{n-1}}^{t} \beta^{t-s}\langle \nabla F(\mathbf{x}_s), \mathbf{u}_s\rangle\right]$$

$$\overset{(II)}{=} \mathbb{E}\left[\beta^{t-t_1+1}\sum_{s=1}^{t_1-1} \beta^{t_1-1-s}\langle \nabla F(\mathbf{x}_{t_1}), \mathbf{u}_{t_1}\rangle + \beta^{t-t_2+1}\sum_{s=t_1}^{t_2-1} \beta^{t_2-1-s}\langle \nabla F(\mathbf{x}_{t_2}), \mathbf{u}_{t_2}\rangle + \cdots\right.$$

$$\left. + \sum_{s=t_{n-1}}^{t} \beta^{t-s}\langle \nabla F(\mathbf{x}_{t_n}), \mathbf{u}_{t_n}\rangle\right]$$

$$\overset{(III)}{=} -\frac{D}{1-\beta}\left(\beta^{t-t_1+1}(1-\beta^{t_1-1})\|\nabla F(\mathbf{x}_{t_1})\| + \beta^{t-t_2+1}(1-\beta^{t_2-t_1})\|\nabla F(\mathbf{x}_{t_2})\| + \cdots\right.$$

$$\left. + (1-\beta^{t-t_{n-1}})\|\nabla F(\mathbf{x}_{t_n})\|\right)$$

$$\overset{(IV)}{\le} -\frac{(1-\beta^t)D}{1-\beta} \inf_{s\in[t]} \|\nabla F(\mathbf{x}_s)\|$$

where $(I)$ is by splitting $t$-shifting stochastic environments into $n$-segments, where $n \in [N]$; $(II)$: under $t_i$-th stationary stochastic environment, we have fixed gradient vector $\nabla F(\mathbf{x}_{t_i})$ and corresponding $\mathbf{u}_{t_i}$; $(III)$ is by setting $\mathbf{u}_{t_i}$ as the best values for each stationary stochastic environment, i.e., $\mathbf{u}_{t_i} := -D \frac{\nabla F(\mathbf{x}_{t_i})}{\|\nabla F(\mathbf{x}_{t_i})\|}$ at $t_i$-th segment; $(IV)$ is due to $\beta^{t-t_1+1}(1 - \beta^{t_1-1}) + \beta^{t-t_2+1}(1 - \beta^{t_2-t_1}) + \cdots + (1 - \beta^{t-t_{n-1}}) = 1 - \beta^t$.

Part 2 can be further re-formulated as

$$
\mathbb{E}\left[\sum_{s=1}^{t} \beta^{t-s}\langle\nabla F(\mathbf{x}_s) - \mathbf{g}_s, -\mathbf{u}_s\rangle\right]
$$

$$
= \mathbb{E}\left[\beta^{t-t_1+1}\sum_{s=1}^{t_1-1}\beta^{t_1-1-s}\langle\nabla F(\mathbf{x}_s) - \mathbf{g}_s, -\mathbf{u}_{t_1}\rangle + \beta^{t-t_2+1}\sum_{s=t_1}^{t_2-1}\beta^{t_2-1-s}\langle\nabla F(\mathbf{x}_s) - \mathbf{g}_s, -\mathbf{u}_{t_2}\rangle + \cdots \right.
$$

$$
\left. + \sum_{s=t_{n-1}}^{t}\beta^{t-s}\langle\nabla F(\mathbf{x}_s) - \mathbf{g}_s, -\mathbf{u}_{t_n}\rangle\right]
$$

$$
\overset{(I)}{\leq} \beta^{t-t_1+1}\sqrt{\mathbb{E}\left[\|\sum_{s=1}^{t_1-1}\beta^{t_1-1-s}(\nabla F(\mathbf{x}_s) - \mathbf{g}_s)\|^2\right]\mathbb{E}\left[\|\mathbf{u}_{t_1}\|^2\right]} + \cdots
$$

$$
+ \sqrt{\mathbb{E}\left[\|\sum_{s=t_{n-1}}^{t}\beta^{t-s}(\nabla F(\mathbf{x}_s) - \mathbf{g}_s)\|^2\right]\mathbb{E}\left[\|\mathbf{u}_{t_n}\|^2\right]}
$$

$$
\overset{(II)}{\leq} D\left(\beta^{t-t_1+1}\sqrt{\sum_{s=1}^{t_1-1}\beta^{2t_1-2-2s}\mathbb{E}\left[\|\nabla F(\mathbf{x}_s) - \mathbf{g}_s)\|^2\right]} + \cdots + \sqrt{\sum_{s=t_{n-1}}^{t}\beta^{2t-2s}\mathbb{E}\left[\|\nabla F(\mathbf{x}_s) - \mathbf{g}_s)\|^2\right]}\right)
$$

$$
\leq \sigma D\left(\beta^{t-t_1+1}\sqrt{\frac{1 - \beta^{2t_1-2}}{1 - \beta^2}} + \beta^{t-t_2+1}\sqrt{\frac{1 - \beta^{2t_2-2t_1}}{1 - \beta^2}} + \cdots + \sqrt{\frac{1 - \beta^{2t-2t_{n-1}}}{1 - \beta^2}}\right)
$$

$$
\leq \left(\beta^{t-t_1+1}\sqrt{1 - \beta^{t_1-1}} + \beta^{t-t_2+1}\sqrt{1 - \beta^{t_2-t_1}} + \cdots + \sqrt{1 - \beta^{t-t_{n-1}}}\right)\frac{\sigma D}{\sqrt{1 - \beta}}
$$

$$
\overset{(III)}{\leq} c_0\frac{\sigma D}{\sqrt{1 - \beta}}
$$

where $(I)$ is due to Cauchy-Schwartz inequality; $(II)$ is by

$$
\mathbb{E}\left[\|\sum_{s=1}^{t_1-1}\beta^{t_1-1-t}(\nabla F(\mathbf{x}_s) - \mathbf{g}_s)\|^2\right]
$$

$$
= \mathbb{E}\left[\sum_{s=1}^{t_1-1}\beta^{2t_1-2-2s}\|\nabla F(\mathbf{x}_s) - \mathbf{g}_s\|^2 + \sum_{1\leq i<j\leq t_1-1}\beta^{t_1-1-i}\beta^{t_1-1-j}\langle\nabla F(\mathbf{x}_i) - \mathbf{g}_i, \nabla F(\mathbf{x}_j) - \mathbf{g}_j\rangle\right]
$$

$$
= \sum_{s=1}^{t_1-1}\beta^{2t_1-2-2s}\mathbb{E}\left[\|\nabla F(\mathbf{x}_s) - \mathbf{g}_s\|^2\right] + \sum_{1\leq i<j\leq t_1-1}\beta^{t_1-1-i}\beta^{t_1-1-j}\langle\nabla F(\mathbf{x}_i) - \mathbb{E}[\mathbf{g}_i], \nabla F(\mathbf{x}_j) - \mathbb{E}[\mathbf{g}_j]\rangle
$$

$$
= \sum_{s=1}^{t_1-1}\beta^{2t_1-2-2s}\mathbb{E}\left[\|\nabla F(\mathbf{x}_s) - \mathbf{g}_s\|^2\right] \qquad \text{Due to i.i.d. sampling.}
$$

$(III)$ is by Definition 4.4.

Part 3 can be further re-formulated as

$$\mathbb{E}\left[\sum_{s=1}^{t}\beta^{t-s}\langle \mathbf{g}_s, \mathbf{z}_s - \mathbf{u}_s\rangle\right]$$

$$= \mathbb{E}\left[\beta^{t-t_1+1}\sum_{s=1}^{t_1-1}\beta^{t_1-1-s}\langle \mathbf{g}_s, \mathbf{z}_s - \mathbf{u}_{t_1}\rangle + \beta^{t-t_2+1}\sum_{s=t_1}^{t_2-1}\beta^{t_2-1-s}\langle \mathbf{g}_s, \mathbf{z}_s - \mathbf{u}_{t_2}\rangle + \cdots\right.$$

$$\left. + \sum_{s=t_{n-1}}^{t}\beta^{t-s}\langle \mathbf{g}_s, \mathbf{z}_s - \mathbf{u}_{t_n}\rangle\right]$$

$$= \mathbb{E}\left[\beta^{t-t_1+1}\text{Regret}_{t_1}^{[\beta]}(\mathbf{u}_{t_1}) + \beta^{t-t_2+1}\text{Regret}_{t_2}^{[\beta]}(\mathbf{u}_{t_2}) + \cdots + \text{Regret}_{t_n}^{[\beta]}(\mathbf{u}_{t_n})\right]$$

$$\leq \left(\beta^{t-t_1+1}\sqrt{1-\beta^{t_1-1}} + \beta^{t-t_2+1}\sqrt{1-\beta^{t_2-t_1}} + \cdots + \sqrt{1-\beta^{t-t_{n-1}}}\right)\sup_{i\in[n]}\frac{\mathbb{E}\left[\text{Regret}_{t_i}^{[\beta]}(\mathbf{u}_{t_i})\right]}{\sqrt{1-\beta^{t_i-t_{i-1}}}}$$

$$\overset{(I)}{\leq} c_0\sup_{i\in[n]}\frac{\mathbb{E}\left[\text{Regret}_{t_i}^{[\beta]}(\mathbf{u}_{t_i})\right]}{\sqrt{1-\beta^{t_i-t_{i-1}}}},$$

where $(I)$ is by Definition 4.4.

Combining the final results of Part 1, Part 2, and Part 3, we have that

$$\mathbb{E}\left[\sum_{s=1}^{t}\beta^{t-s}\left(F(\mathbf{x}_s) - F(\mathbf{x}_{s-1})\right)\right] \leq -\frac{(1-\beta^t)D}{1-\beta}\inf_{s\in[t]}\|\nabla F(\mathbf{x}_s)\| + c_0\frac{\sigma D}{\sqrt{1-\beta}} + c_0\sup_{i\in[n]}\frac{\mathbb{E}\left[\text{Regret}_{t_i}^{[\beta]}(\mathbf{u}_{t_i})\right]}{\sqrt{1-\beta^{t_i-t_{i-1}}}},$$

which concludes the proof. $\qquad\square$

**Theorem 4.6** (Static stochastic environment discounted-to-nonconvex conversion). *Supposing that $F$ satisfies Assumption 2.1, the Algorithm 1 guarantees*

$$\mathbb{E}_\tau\|\mathbb{E}_{\widetilde{\mathbf{X}}_\tau}\nabla F(\widetilde{\mathbf{X}}_\tau)\| \leq \frac{c_0}{DT}\left(\beta_1\mathbb{E}\left[\sup_{i\in[N]}\frac{\mathbb{E}\left[Regret_{t_i}^{[\beta]}(\mathbf{u}_{t_i})\right]}{\sqrt{1-\beta_1^{t_i-t_{i-1}}}}\right] + (1-\beta_1)\sum_{t=1}^{T}\mathbb{E}\left[\sup_{i\in[n]}\frac{\mathbb{E}\left[Regret_{t_i}^{[\beta]}(\mathbf{u}_{t_i})\right]}{\sqrt{1-\beta_1^{t_i-t_{i-1}}}}\right]\right)$$

$$+ \frac{\Delta}{DT} + \frac{c_0\sigma}{T\sqrt{1-\beta_1}} + c_0\sigma\sqrt{1-\beta_1}.$$

*where $\widetilde{\mathbf{X}}_\tau$ is distributed over $\{\widetilde{\mathbf{x}}_t\}_{t=1}^{T}$ as* $\mathbb{P}(\tau = t) = \begin{cases} \frac{1-\beta_1^t}{T} & \text{for } t = 1, \cdots, T-1, \\ \frac{1}{1-\beta_1}\frac{1-\beta_1^T}{T} & \text{for } t = T. \end{cases}$

The proof of Theorem 4.6 is omitted since, given Lemma 4.5, it simply replaces our supremum-type discounted regret with vanilla discounted regret in the existing result (Lemma 1 of Ahn et al. (2024a)).

## A.2   PROOFS FOR SECTION 5

**Lemma 5.1.** *When $\beta_1 \leq \beta_2 \leq \sqrt{\beta_1}$, Adam-FTRL satisfies $\|\mathbf{z}_t\|^2 \leq \frac{4\eta^2}{1-\beta_1}$.*

*Proof.*

$$\|\mathbf{z}_t\|^2 = \eta^2\frac{1-\beta_2^{t-1}}{(1-\beta_1^{t-1})^2}\frac{\|(1-\beta_1)\sum_{s=1}^{t-1}\beta_1^{t-1-s}\mathbf{v}_s\|^2}{(1-\beta_2)\sum_{s=1}^{t-1}\beta_2^{t-1-s}\|\mathbf{v}_s\|^2}$$

$$\overset{(i)}{\leq} \eta^2\frac{1-\beta_2^{t-1}}{(1-\beta_1^{t-1})^2}\frac{(1-\beta_1)^2(\sum_{s=1}^{t-1}\beta_2^{t-1-s}\|\mathbf{v}_s\|^2)(\sum_{s=1}^{t-1}\frac{\beta_1^{2t-2-2s}}{\beta_2^{t-1-s}})}{(1-\beta_2)\sum_{s=1}^{t-1}\beta_2^{t-1-s}\|\mathbf{v}_s\|^2}$$

$$\leq \eta^2\frac{1-\beta_2^{t-1}}{(1-\beta_1^{t-1})^2}\frac{(1-\beta_1)^2}{(1-\beta_2)}\sum_{s=1}^{t-1}\left(\frac{\beta_1^2}{\beta_2}\right)^{t-1-s}$$

where $(i)$ is by the weighted Cauchy–Schwarz inequality.

Choosing $\beta_1 \leq \beta_2 \leq \sqrt{\beta_1}$, we have

$$\frac{(1-\beta_1)^2}{(1-\beta_2)}\sum_{s=1}^{t-1}\left(\frac{\beta_1^2}{\beta_2}\right)^{t-1-s} = \frac{(1-\beta_1)^2}{(1-\beta_2)}\frac{1-\left(\frac{\beta_1^2}{\beta_2}\right)^{t-1}}{1-\frac{\beta_1^2}{\beta_2}} \leq \frac{(1-\beta_1)^2}{(1-\beta_2)^2} \leq \frac{(1-\sqrt{\beta_1})^2(1+\sqrt{\beta_2})^2}{(1-\sqrt{\beta_1})^2} \leq 4,$$

and $\frac{1-\beta_2^{t-1}}{(1-\beta_1^{t-1})^2} \leq \frac{1}{1-\beta_1^{t-1}}$. Then, it suffices to have $\|\mathbf{z}_t\|^2 \leq \frac{4\eta^2}{1-\beta_1^{t-1}} \leq \frac{4\eta^2}{1-\beta_1}$. $\qquad\square$

### A.2.1 PROOFS FOR SECTION 5.2

**Theorem 5.2** (Discounted regret of Adam-FTRL). *Assume Assumption 4.2 holds. For a $\beta$-discounted loss sequence $\beta_1^{t-1}\mathbf{v}_1, \cdots, \beta_1^{t-t}\mathbf{v}_t$, comparator $\mathbf{u}_t \in \mathcal{D}$, i.e., $\|\mathbf{u}_t\| \leq D$, Adam-FTRL guarantees $\mathbb{E}\left[Regret_t^{[\beta]}(\mathbf{u}_t)\right] = 3\sqrt{1+2c_1}D\sqrt{\frac{1-\beta_1^t}{1-\beta_1}}\sqrt{Tr(\boldsymbol{\Sigma}_t)+\|\boldsymbol{\mu}_t\|^2}$ when the learning rate is chosen as $\eta = \sqrt{1-\beta_1}D$.*

*Proof.* We begin with

$$\mathbb{E}\left[\sum_{s=1}^t \langle \beta_1^{t-s}\mathbf{v}_s, \mathbf{z}_s - \mathbf{u}\rangle\right] = \underbrace{\mathbb{E}\left[\sum_{s=1}^t \langle \beta_1^{t-s}\mathbf{v}_s, \mathbf{z}_s\rangle\right]}_{\text{Part 1}} + \underbrace{\mathbb{E}\left[\sum_{s=1}^t \langle \beta_1^{t-s}\mathbf{v}_s, -\mathbf{u}\rangle\right]}_{\text{Part 2}}$$

- **Part 1**

$$\mathbb{E}\left[\sum_{s=1}^t \langle \beta_1^{t-s}\mathbf{v}_s, \mathbf{z}_s\rangle\right] \overset{(i)}{\leq} \sum_{s=1}^t \beta_1^{t-s}\sqrt{\mathbb{E}\left[\|\mathbf{v}_s\|^2\right]\mathbb{E}\left[\|\mathbf{z}_s\|^2\right]}$$

$$\overset{(ii)}{\leq} \frac{2\eta}{\sqrt{1-\beta_1}}\sqrt{\text{Tr}(\boldsymbol{\Sigma}_t)+\|\boldsymbol{\mu}_t\|^2}\frac{1-\beta_1^t}{1-\beta_1}$$

$$\leq 2\eta\frac{1-\beta_1^t}{(1-\beta_1)^{\frac{3}{2}}}\sqrt{\text{Tr}(\boldsymbol{\Sigma}_t)+\|\boldsymbol{\mu}_t\|^2}$$

where $(i)$ is by Cauchy-Schwarz Inequality; $(ii)$ is by Lemma 5.1.

- **Part 2**

$$\mathbb{E}\left[\sum_{s=1}^t \langle \beta_1^{t-s}\mathbf{v}_s, -\mathbf{u}\rangle\right] \leq D\mathbb{E}\left[\|\sum_{s=1}^t \beta_1^{t-s}\mathbf{v}_s\|\right]$$

$$\leq D\sqrt{\sum_{s=1}^t \beta_1^{2t-2s}\mathbb{E}\left[\|\mathbf{v}_s\|^2\right] + 2\sum_{1\leq i<j\leq t}\beta_1^{t-i}\beta_1^{t-j}\langle\mathbb{E}\left[\mathbf{v}_i\right],\mathbb{E}\left[\mathbf{v}_j\right]\rangle}$$

$$\leq D\sqrt{\text{Tr}(\boldsymbol{\Sigma}_t)+\|\boldsymbol{\mu}_t\|^2}\sqrt{\frac{1-\beta_1^{2t}}{1-\beta_1^2}+2c_1\frac{(1-\beta_1^t)^2}{1-\beta_1}}$$

$$\leq \sqrt{1+2c_1}D\sqrt{\frac{1-\beta_1^t}{1-\beta_1}}\sqrt{\text{Tr}(\boldsymbol{\Sigma}_t)+\|\boldsymbol{\mu}_t\|^2}$$

Then, summing over Part 1 and Part 2 gives

$$\sum_{s=1}^t \langle \beta_1^{t-s}\mathbf{v}_s, \mathbf{z}_s - \mathbf{u}\rangle \leq 2\eta\frac{1-\beta_1^t}{(1-\beta_1)^{\frac{3}{2}}}\sqrt{\text{Tr}(\boldsymbol{\Sigma}_t)+\|\boldsymbol{\mu}_t\|^2} + \sqrt{1+2c_1}D\sqrt{\frac{1-\beta_1^t}{1-\beta_1}}\sqrt{\text{Tr}(\boldsymbol{\Sigma}_t)+\|\boldsymbol{\mu}_t\|^2}$$

$$\leq 3\sqrt{1+2c_1}D\sqrt{\frac{1-\beta_1^t}{1-\beta_1}}\sqrt{\text{Tr}(\boldsymbol{\Sigma}_t)+\|\boldsymbol{\mu}_t\|^2}$$

where the last inequality is by setting $\eta = \sqrt{1-\beta_1}D \geq \frac{1-\beta_1}{\sqrt{1-\beta_1^t}}D$ and $2 + \sqrt{1+2c_1} \leq 3\sqrt{1+2c_1}$, which concludes the proof. $\qquad\square$

### A.3 Proofs for Section 6

**Theorem 6.1.** *Assume that $F$ satisfies Assumption 2.1 and consider any $\lambda > 0$. Algorithm 1, when selecting $\mathcal{A}$ as Adam-FTRL, guarantees*

$$\mathbb{E}_\tau \left[ \|\nabla F(\widetilde{\mathbf{x}})\|^{[\lambda]} \right] \leq (2 + 4c_0\sqrt{1+2c_1})\epsilon$$

*when the parameters are chosen as $T = \max\left( 8\Delta\lambda^{\frac{1}{2}}(G+\sigma)^2\epsilon^{-\frac{7}{2}}, 4c_0\sqrt{1+2c_1}(G+\sigma)^2\epsilon^{-2} \right)$, $\beta_1 = 1 - \frac{\epsilon^2}{(G+\sigma)^2}$, and $D = \frac{1}{8}(1-\beta_1)\lambda^{-\frac{1}{2}}\epsilon^{\frac{1}{2}}$. I.e., $\widetilde{\mathbf{x}}$ is a $(\lambda, (2+4c_0\sqrt{1+2c_1})\epsilon)$-stationary point.*

*Proof.* By Definition 2.2, it has

$$\mathbb{E}_\tau \left[ \|\nabla F(\widetilde{\mathbf{x}})\|^{[\lambda]} \right] \leq \mathbb{E}_\tau \left[ \|\mathbb{E}_{\widetilde{\mathbf{X}}_\tau} \nabla F(\widetilde{\mathbf{X}}_\tau)\| + \lambda \mathbb{E}_{\widetilde{\mathbf{X}}_\tau} \|\widetilde{\mathbf{X}}_\tau - \widetilde{\mathbf{x}}\|^2 \right],$$

$\widetilde{\mathbf{X}}_\tau$ is distributed over $\{\widetilde{\mathbf{x}}_t\}_{t=1}^T$ as $\mathbb{P}(\tau = t) = \begin{cases} \frac{1-\beta^t}{T} & \text{for } t = 1, \cdots, T-1, \\ \frac{1}{1-\beta}\frac{1-\beta^T}{T} & \text{for } t = T. \end{cases}$

**Fisrtly, we deal with $\lambda\mathbb{E}_\tau \left[ \mathbb{E}_{\widetilde{\mathbf{X}}_\tau} \|\widetilde{\mathbf{X}}_\tau - \widetilde{\mathbf{x}}\|^2 \right]$.**

By Lemma 8 of Ahn et al. (2024a), we have $\mathbb{E}_\tau \left[ \mathbb{E}_{\widetilde{\mathbf{X}}_\tau} \|\widetilde{\mathbf{X}}_\tau - \widetilde{\mathbf{x}}\|^2 \right] \leq \frac{4\beta_1}{(1-\beta_1)^2 T}\mathbb{E}\left[ \sum_{t=1}^T \|\mathbf{z}_t\|^2 \right]$. Since $\mathbb{E}\left[ \sum_{t=1}^T \|\mathbf{z}_t\|^2 \right] = 16D^2 T$ (by Lemma 5.1 and Theorem 5.2), it suffices to have $\lambda\mathbb{E}_{t\in[T]}\mathbb{E}_{\widetilde{\mathbf{X}}_\tau} \|\widetilde{\mathbf{X}}_\tau - \widetilde{\mathbf{x}}\|^2 \leq \frac{64\lambda D^2}{(1-\beta_1)^2}$.

By choosing $D = \frac{1}{8}(1-\beta_1)\lambda^{-\frac{1}{2}}\epsilon^{\frac{1}{2}}$, it guarantees $\lambda\mathbb{E}_\tau \left[ \mathbb{E}_{\widetilde{\mathbf{X}}_\tau} \|\widetilde{\mathbf{X}}_\tau - \widetilde{\mathbf{x}}\|^2 \right] = \epsilon$.

**Secondly, we deal with $\mathbb{E}_\tau \|\mathbb{E}_{\widetilde{\mathbf{X}}_\tau} \nabla F(\widetilde{\mathbf{X}}_\tau)\|$.**

Substituting the supremum-type discounted regret in the regret conversion bound of Theorem 4.6 with the regret bound of Theorem 5.2 gives

$$\mathbb{E}_\tau \|\mathbb{E}_{\widetilde{\mathbf{X}}_\tau} \nabla F(\widetilde{\mathbf{X}}_\tau)\|$$
$$\leq \frac{\Delta}{DT} + \frac{c_0\sigma}{T\sqrt{1-\beta_1}} + c_0\sigma\sqrt{1-\beta_1} + \frac{3c_0\sqrt{1+2c_1}(G+\sigma)}{T\sqrt{1-\beta_1}} + 3c_0\sqrt{1+2c_1}(G+\sigma)\sqrt{1-\beta_1}$$
$$\overset{(i)}{=} \frac{8\Delta\lambda^{\frac{1}{2}}}{(1-\beta_1)\epsilon^{\frac{1}{2}}T} + \frac{c_0\sigma}{T\sqrt{1-\beta_1}} + c_0\sigma\sqrt{1-\beta_1} + \frac{3c_0\sqrt{1+2c_1}(G+\sigma)}{T\sqrt{1-\beta_1}} + 3c_0\sqrt{1+2c_1}(G+\sigma)\sqrt{1-\beta_1}$$
$$\overset{(ii)}{=} \frac{8\Delta\lambda^{\frac{1}{2}}(G+\sigma)^2}{\epsilon^{\frac{5}{2}}T} + \frac{c_0(G+\sigma)\sigma}{\epsilon T} + c_0\epsilon + \frac{3c_0\sqrt{1+2c_1}(G+\sigma)^2}{\epsilon T} + 3c_0\sqrt{1+2c_1}\epsilon$$
$$\leq \frac{8\Delta\lambda^{\frac{1}{2}}(G+\sigma)^2}{\epsilon^{\frac{5}{2}}T} + \frac{4c_0\sqrt{1+2c_1}(G+\sigma)^2}{\epsilon T} + 4c_0\sqrt{1+2c_1}\epsilon$$
$$\leq (2 + 4c_0\sqrt{1+2c_1})\epsilon$$

where $(i)$ is by choosing $D = \frac{1}{8}(1-\beta_1)\lambda^{-\frac{1}{2}}\epsilon^{\frac{1}{2}}$; $(ii)$ is by choosing $\beta_1 = 1 - \frac{\epsilon^2}{(G+\sigma)^2}$; the last equality is by choosing $T = \max\left( 8\Delta\lambda^{\frac{1}{2}}(G+\sigma)^2\epsilon^{-\frac{7}{2}}, 4c_0\sqrt{1+2c_1}(G+\sigma)^2\epsilon^{-2} \right)$, which concludes the proof. $\qquad\square$

## B SMOOTH-DRIFT ENVIRONMENT IS NO HARDER THAN A-FEW-SEGMENTS ENVIRONMENT FOR ANY ONLINE LEARNERS

### B.1 SETUP AND NOTATION

Let $\{\mathbf{z}_t\}_{t=1}^T$ be the learner's decisions and $\mathbf{u}$ a fixed comparator. At round $t$ the environment produces a cost vector $\mathbf{v}_t \in \mathbb{R}^d$ with mean $\boldsymbol{\mu}_t := \mathbb{E}[\mathbf{v}_t \mid \mathcal{F}_{t-1}]$, where $\{\mathcal{F}_t\}$ is the natural filtration.

**Partition of environment and static segments.** A environmental partition $\Pi = \{I_k = [a_k : b_k]\}_{k=1}^K$ consists of disjoint consecutive intervals covering $\{1, \ldots, T\}$. We write $\Delta(I) = |I| = b - a + 1$ where $I = [a : b]$ denotes a specific interval/segment. Given $\Pi$, we consider a piecewise-static reference environment $\{\mathbf{v}_t^\Pi\}_{t \in [T]}$ such that for any segment $I = [a : b] \in \Pi$, $\boldsymbol{\mu}_t^\Pi := \mathbb{E}[\mathbf{v}_t^\Pi \mid \mathcal{F}_{t-1}]$ is constant in $t$ over $I$.

### B.2 PART I: ANY PARTITION CASE

**Theorem B.1** (Discounted regret coupling). *For the same online learning algorithm,*

$$\mathbb{E}\left[ Regret_{1:T}^{[\beta]}(\mathbf{u}) \right] \le 2 \sup_{I \in \Pi} \left( \mathbb{E}\left[ Regret_I^{[\beta],\Pi}(\mathbf{u}) \right] + \frac{6GD}{1-\beta} \right),$$

*where for $I = [a : b]$ and $\beta \in (0, 1)$, $Regret_I^{[\beta],\Pi}(\mathbf{u}) := \sum_{t \in I} \beta^{b^I - t} \langle \mathbf{v}_t^\Pi, \mathbf{z}_t - \mathbf{u} \rangle$.*

*Proof.* We start from

$$\mathbb{E}\left[ \mathrm{Regret}_{1:T}^{[\beta]}(\mathbf{u}) \right] = \mathbb{E}\left[ \sum_{t=1}^T \beta^{T-t} \langle \mathbf{v}_t, \mathbf{z}_t - \mathbf{u} \rangle \right]$$

$$= \mathbb{E}\left[ \sum_{I \in \Pi} \beta^{T-b^I} \sum_{t \in I} \beta^{b^I - t} \left( \langle \mathbf{v}_t^\Pi, \mathbf{z}_t - \mathbf{u} \rangle + \langle \mathbf{v}_t - \mathbf{v}_t^\Pi, \mathbf{z}_t - \mathbf{u} \rangle \right) \right]$$

$$\le \left( \sum_{I \in \Pi} \beta^{T-b^I} \right) \left( \sup_{I \in \Pi} \mathbb{E}\left[ \mathrm{Regret}_I^{[\beta],\Pi}(\mathbf{u}) \right] \right) + \mathbb{E}\left[ \sum_{I \in \Pi} \beta^{T-b^I} \sum_{t \in I} \beta^{b^I - t} \langle \mathbf{v}_t - \mathbf{v}_t^\Pi, \mathbf{z}_t - \mathbf{u} \rangle \right].$$

$$\tag{2}$$

Regarding a segment $I$,

$$\sum_{t \in I} \mathbb{E}\left[ \beta^{b^I - t} \langle \mathbf{v}_t - \mathbf{v}_t^\Pi, \mathbf{z}_t - \mathbf{u} \rangle \right] = \sum_{t \in I} \mathbb{E}\left[ \mathbb{E}\left[ \beta^{b^I - t} \langle \mathbf{v}_t - \mathbf{v}_t^\Pi, \mathbf{z}_t - \mathbf{u} \rangle \mid \mathcal{F}_{t-1} \right] \right] \quad \text{(tower property)}$$

$$= \sum_{t \in I} \mathbb{E}\left[ \beta^{b^I - t} \left\langle \mathbb{E}\left[ \mathbf{v}_t - \mathbf{v}_t^\Pi \mid \mathcal{F}_{t-1} \right], \mathbf{z}_t - \mathbf{u} \right\rangle \right] \quad (\mathbf{z}_t \text{ is } \mathcal{F}_{t-1}\text{-measurable})$$

$$= \sum_{t \in I} \mathbb{E}\left[ \beta^{b^I - t} \langle \boldsymbol{\mu}_t - \boldsymbol{\mu}_t^\Pi, \mathbf{z}_t - \mathbf{u} \rangle \right]$$

$$\le \sum_{t \in I} \beta^{b^I - t} \| \boldsymbol{\mu}_t - \boldsymbol{\mu}_t^\Pi \| \| \mathbf{z}_t - \mathbf{u} \|$$

$$\overset{(I)}{\le} 6GD \frac{1 - \beta^{b^I - a^I + 1}}{1 - \beta}$$

$$= 6GD \frac{1 - \beta^{\Delta(I)}}{1 - \beta} \tag{3}$$

where $(I)$ is by: (a).Under the setting of Adam-FTRL, we have $\| \mathbf{z}_t - \mathbf{u} \| \le 3D$; (b). $\| \boldsymbol{\mu} \| \le G$.

Substituting the result of equation 3 into equation 2,

$$\mathbb{E}\left[ \mathrm{Regret}_{1:T}^{[\beta]}(\mathbf{u}) \right] \le \left( \sum_{I \in \Pi} \beta^{T-b^I} \right) \sup_{I \in \Pi} \left( \mathbb{E}\left[ \mathrm{Regret}_I^{[\beta],\Pi}(\mathbf{u}) \right] + 6GD \frac{1 - \beta^{\Delta(I)}}{1 - \beta} \right)$$

Note that the above inequality holds for any partition $\Pi$; then we choose a partition $\Pi$ consisting of two segments, which further gives

$$\mathbb{E}\left[\text{Regret}_{1:T}^{[\beta]}(\mathbf{u})\right] \leq 2\sup_{I\in\Pi}\left(\mathbb{E}\left[\text{Regret}_I^{[\beta],\Pi}(\mathbf{u})\right] + \frac{6GD}{1-\beta}\right)$$

$\square$

