# OpenReview forum: "The Role of Stochastic Environments in Enabling Adam"
_ICLR.cc/2026/Conference — Submitted to ICLR 2026_

### Official Review · Reviewer_ZT1z · 2025-10-26

**Soundness:** 3
**Presentation:** 2
**Contribution:** 3
**Rating:** 6
**Confidence:** 2

**Summary:**

This paper investigates the discounted-to-nonconvex framework under shifting stochastic environments. Furthermore, it introduces an online learning algorithm, Adam-FTRL, which incorporates both momentum and bias correction mechanisms.

**Strengths:**

1. This paper presents a detailed theoretical analysis of the proposed methods, particularly the Adam-FTRL algorithm, which integrates both momentum and bias correction mechanisms.

2. In addition, the authors demonstrate optimal iteration complexity for nonconvex optimization.

**Weaknesses:**

Although the paper provides sufficient theoretical analysis and introduces new methods, it lacks numerical experiments to validate their performance.

Compared with some existing works (Li et al., 2024; Wang et al., 2024), the G-Lipschitz assumption may be too strong, and the final results rely on the constants  $c_0$ and $c_1$.

**Questions:**

Despite its strengths, I have some concerns about the novelty of the work. The discounted-to-nonconvex conversion framework has already been extensively studied in prior research (Zhang & Cutkosky, 2024). In this paper, the conversion algorithm closely follows that framework, with the main modification being the reformulation of the online algorithm to recover the original Adam. Furthermore, many of the analytical techniques rely heavily on lemmas from existing works.

Could the authors further clarify their contributions regarding the shifting stochastic environment and the robust convergence of Adam-FTRL? In Definition 4.1, if u_t is set as the average of gradients and v_t as the stochastic estimate, the subsequent assumptions appear fairly standard in stochastic optimization. Am I missing something here?

---

> ### Author Response · Authors · 2025-11-16
>
> We thank the reviewer ZT1z for the positive feedback. We address each concern in detail below.
>
>
> ---
>
>
> $\textbf{On numerical experiments}$
>
> Our primary goal in this paper is to develop a theoretical framework that more faithfully explains the empirical success of plain Adam, rather than proposing a new optimizer that seeks empirical superiority.
> Because our contribution is fundamentally theoretical, removing restrictive assumptions and analyzing the exact Adam update, our focus is on establishing the mathematical foundations.
>
> As such, the most impactful next step is to continue bridging the theory–practice gap at the analytical level, for example, by formally relating smooth-drift environments and a-few-segment environments via regret analysis. Additional numerical experiments, while valuable, lie outside the immediate scope of this submission.
>
> ---
>
> $\textbf{G-Lipschitz assumption}$
> * The G-Lipschitz assumption is standard in the nonsmooth nonconvex optimization literature [1,2].
> * As noted in the manuscript, the optimality notion, $(\lambda, \rho)$-stationarity, has the desirable property: whenever the underlying objective is smooth, $(\lambda, \rho)$-stationarity can be reduced to standard first-order stationarity. Consequently, the guarantees obtained under the G-Lipschitz setting immediately translate into convergence guarantees for smooth nonconvex optimization [3].
>
> ---
>
>
> $\textbf{Novelty concerns: differences and contributions}$
> * $\textbf{Algorithmic contributions}$
>   * Dropping the clipping operation: Clipping introduces an additional problem-dependent hyperparameter (the clipping threshold). Removing it eliminates the need for two-dimensional hyperparameter tuning and makes the analysis better aligned with how Adam is used in practice.
>   * Analyzing the exact Adam update:  Most prior theoretical works analyze heavily modified versions of Adam. We instead analyze the true update rule, which is substantially more difficult. This reveals properties of Adam that remain hidden in surrogate formulations.
> * $\textbf{Analytical contributions}$
>   This part is subtle and, in our view, the most significant. There are choices between: (i) an environment that is fully realistic but only analyzable by oversimplified algorithms, or (ii) an environment that is stylized but analytically compatible with the practical Adam update. Our work deliberately follows (ii). We view this as a conceptual contribution\shift rather than a mere technical workaround.
>   * We introduce a new structured environment model that is compatible with Adam’s dynamics and enables robust convergence of Adam-FTRL without clipping or restrictive hyperparameter couplings.
>
>   This is not a minor modification of earlier frameworks. Rather, it is a different modeling choice.
>
> ---
>
>
> $\textbf{Clarification regarding Definition 4.1}$
>
> The reviewer is correct that the objects in Definition 4.1 resemble standard notions in stochastic optimization: $\mu_{t}$ can be viewed as the underlying $\nabla F(x_{t})$; $v_{t}$ represents a stochastic estimate, i.e., $v_{t} \rightarrow g(x_{t})$.
>
> This is intentional: the purpose is to connect our shifting-environment model with classical assumptions.
>
> ---
>
> [1] Cutkosky, Ashok, Harsh Mehta, and Francesco Orabona. "Optimal stochastic non-smooth non-convex optimization through online-to-non-convex conversion." International Conference on Machine Learning. PMLR, 2023.
>
> [2] Zhang, Jingzhao, et al. "Complexity of finding stationary points of nonconvex nonsmooth functions." International Conference on Machine Learning. PMLR, 2020.
>
> [3] Ahn, Kwangjun, and Ashok Cutkosky. "Adam with model exponential moving average is effective for nonconvex optimization." Advances in Neural Information Processing Systems 37 (2024): 94909-94933.

---

> ### Author Response · Authors · 2025-11-23
>
> This message is intended solely as a gentle reminder for Reviewer ZT1z. We sincerely apologize for the continued email notifications.
>
> We truly hope for a productive and constructive discussion, and kindly expect that any exchanges take place here.
>
> Thank you for your understanding.
>
> Best regards,
>
> Authors of submission 21021

---

> > ### Comment · Reviewer_ZT1z · 2025-11-27
> >
> > I am satisfied with the response and choose to keep my positive score.

---

### Official Review · Reviewer_G6VC · 2025-10-28

**Soundness:** 2
**Presentation:** 2
**Contribution:** 2
**Rating:** 2
**Confidence:** 4

**Summary:**

This paper provides a refined discounted-to-nonconvex analysis for Adam in the context of non-smooth and non-convex optimization. To derive a better regret bound for Adam, the authors propose an online learning with shifting stochastic environments framework. Then, they introduce Adam-FTRL, an online algorithm variant of Adam, which can transfer the non-smooth/non-convex problem to an online learning problem. They further prove the competitive discounted regret bounds without clipping or unrealistic parameter couplings.

**Strengths:**

- The major novelty of this paper lies in introducing the shifting stochastic environments framework with some specific structures, and proposing Adam-FTRL without clipping. The former one may help to derive a better regret bound for Adam. The latter one is a more realistic form for Adam.

- The authors further provide some theoretical regret bounds to illustrate the effectiveness of Adam-FTRL.

**Weaknesses:**

I have some concerns as follows.

- The environmental constants $c_0$ and $c_1$ look very strange, without any obvious explanation or intuition. In my view, $c_1$ is the ratio between the mean and covariance of the cost vector ${\bm v}_t$, while $c_0$, is only a formula dependent on $\beta_1$, which emerges during the regret analysis. The authors claim that "The environmental constants $c_0$ and $c_1$ provide valuable theoretical
insight into the role of stochasticity and shifting complexity.", without convincing evidence.

- The whole paper is very similar to [1], particularly the regret analysis. The only major difference, in my view, is to drop the clipping of Adam. However, I do not see any obvious advantages brought by the Adam-FTRL without clipping. In addition, I think dropping the clipping would not essentially affect the regret analysis. If the analysis is essentially different, I suggest that the authors provide a detailed discussion.

- There are also some presentation problems. For example, Lemma 4.5 is established based on using $g_t$, the stochastic gradient, as the cost vector, which is not clearly presented.

[1] Kwangjun Ahn and Ashok Cutkosky. Adam with model exponential moving average is effective for nonconvex optimization. arXiv preprint arXiv:2405.18199, 2024.

**Questions:**

- What's the major difference between this paper to [1]? If the major difference lies in incorporating the corrective term and dropping the clipping, it would be better to explain the additional challenge brought by these changes.

- Are there any intuition for $c_0$ and $c_1$? In my view, the two terms are just the mathematical formulas. I suggest the authors to provide more explanation.

---

> ### Author Response · Authors · 2025-11-15
>
> We thank the reviewer G6VC for the feedback. We address each concern in detail below.
>
> ---
>
> $\textbf{On dropping the clipping component of Adam}$
>
> Thank you for raising this important point. The role of clipping in Adam is under-discussed, and we clarify our contributions below.
>
> * (i) $\textbf{Dropping clipping at the algorithmic level}$
>   * Removing clipping eliminates an additional tunable hyperparameter (the clipping threshold).
>   * In practice, this threshold is highly problem-dependent. Compared with standard Adam, which typically tunes only the learning rate, “Adam with clipping” requires a $\textbf{two-dimensional grid search}$ over learning rate and clipping threshold. Thus, eliminating clipping reduces tuning cost.
>
> * (ii) $\textbf{Dropping clipping at the analytical level}$
>   This aspect is more subtle and more significant for theory: clipping greatly simplifies previous analyses.
>   * Clipping arbitrarily controls the increment of the update, which makes it easier to bound the effective learning-rate scaling,
>   *  but simultaneously suppresses the influence of Adam’s internal hyperparameters $\beta_{1}, \beta_{2}$.
>
>   In contrast, we analyze the exact Adam update without clipping, preserving the real behavior used in practice. This is technically harder: the stability of the adaptive step scales must be shown through the dynamics of EMA terms rather than by enforcing hard truncation.
>
>   Therefore, dropping clipping is not only an algorithmic simplification but also a major analytical challenge that distinguishes our work from prior analyses.
>
>
> ---
>
> $\textbf{Differences in regret analysis and contributions}$
>
> We appreciate the reviewer’s request for a clearer comparison. Contrary to the impression that our analysis is similar to [1], there are several structural differences.
>
> * (i) $\textbf{Existing analyses heavily count on unaligned algorithmic configuration}$
>   The regret guarantee in [1] is stated, but the proof is omitted. Based on the structure of the bound, we believe its analysis strongly relies on the earlier work [2], which fundamentally assumes:
>   * unrealistic clipping operation, very restrictive couplings of $\beta_{1}$ and $\beta_{2}$, and omits critical components of Adam.
>
>   As discussed in the manuscript, extending those analyses to the exact Adam update appears extremely challenging, and to our knowledge, no prior work fully resolves this difficulty.
>
> * (ii) $\textbf{Our analytical framework is structurally different}$
>   Our contribution is to introduce a new analytical framework that trades a tighter regret bound for a structured stochastic environment that is compatible with the full Adam update.
>   The core logic of our derivation is fundamentally different. For example, our discounted regret bound contains the additional term $\sqrt{1 - \beta_{1}^{t}}$, and the bounding steps rely on controlling shifting stochastic environments rather than simple clipping-based truncation.
> * (iii) $\textbf{Summary}$ Thus, the only shared high-level theme is “analyzing Adam via FTRL,” but the techniques and intermediate guarantees differ substantially.
>
>
> ---
>
> $\textbf{Intuition for the environmental constants}$ $c_{0}$ $\textbf{and}$ $c_{1}$
>
> Thank you for highlighting the need for more intuition. We clarify their roles below.
>
> * Intuition of $c_{1}$.
>   At each time step $t$, the distribution of the stochastic gradient has mean $\mu_{t}$​ and covariance $\Sigma_{t}$. The constant $c_{1}$​ bounds a normalized signal-to-noise ratio for this distribution.
>   * Large mini-batches $\rightarrow$ small covariance $\rightarrow$ higher signal-to-noise $\rightarrow$ larger $c_{1}$.
>   * Small mini-batches $\rightarrow$ noisier gradients $\rightarrow$ smaller $c_{1}$.
>
>   In the limit of full-batch gradients, $c_{1}$​ approaches $1/(1-\beta_{1})$. Hence, $c_{1}$​ summarizes the intrinsic stochasticity of the environment.
>
> * Intuition of $c_{0}$.
>   The constant $c_{0}$​ characterizes the frequency of distributional changes across training. Consider $N$ segments for $T$ iterations, various $c_{0}$ correspond to various segmentation conditions. While, $N \equiv T$ push $c_{0}$ to worst case as $1/\sqrt{1 - \beta_{1}}$.
>
> ---
>
>
> $\textbf{Presentation issues}$ We will revise the manuscript to explicitly clarify the use of $g_{t}$ as the cost vector.
>
> ---
>
> [1] Ahn, Kwangjun, and Ashok Cutkosky. "Adam with model exponential moving average is effective for nonconvex optimization." Advances in Neural Information Processing Systems 37 (2024): 94909-94933.
>
> [2] Ahn, Kwangjun, et al. "Understanding Adam optimizer via online learning of updates: Adam is FTRL in disguise." arXiv preprint arXiv:2402.01567 (2024).

---

> > ### Comment · Reviewer_G6VC · 2025-11-24
> >
> > 1. Can you provide some more details about this sentence: "As discussed in the manuscript, extending those analyses to the exact Adam update appears extremely challenging, and to our knowledge, no prior work fully resolves this difficulty."
> >
> > 2. Can you provide some more details about this sentence: "The core logic of our derivation is fundamentally different. For example, our discounted regret bound contains the additional term $\sqrt{1 - \beta_{1}^{t}}$, and the bounding steps rely on controlling shifting stochastic environments rather than simple clipping-based truncation."
> >
> > It would be better to explain with some detailed mathematical formula.
> >
> > 3. The roles of $c_0$ and $c_1$ are still not very clear. For example, what are these sentences mean? "The constant $c_{0}$​ characterizes the frequency of distributional changes across training. Consider $N$ segments for $T$ iterations, various $c_{0}$ correspond to various segmentation conditions."

---

> > > ### Author Response · Authors · 2025-11-25
> > >
> > > We thank the reviewer for the follow-up questions and appreciate the opportunity to clarify these issues with more detail. Below, we address each point carefully.
> > >
> > > ---
> > >
> > > $\textbf{Point 1}$
> > >
> > > The line of work in [1,2,3] is highly relevant to our study, as it establishes a bridge between Adam-type methods and the online learning (FTRL) framework. However, none of these studies analyzes the exact Adam update as used in practice, and extending their techniques to this exact form is technically nontrivial for several concrete reasons:
> > >   * The regret bound in [1, 2] relies on the rigid coupling condition $\beta_{1}^{2} = \beta_{2}$, demonstrated as proof of  Theorem B.2 in [1] and Theorem 9 in [2]. Once this coupling is removed, the key algebraic cancellations disappear, and the classical proof strategy no longer applies. Addressing the general case therefore requires a fundamentally new analytical route.
> > >   * Those works rely on a clipped version of the incremental term $z_{t}$. In contrast, our analysis directly incorporates the bias-correction terms used in Adam and bounds the true update through a carefully designed but realistically admissible coupling of $\beta_{1}$​ and $\beta_{2}$ ​ (see Lemma 5.1).
> > > * These previous surrogate formulations obscure important aspects of Adam’s true design. Our analysis recovers several of these hidden principles by working directly with the exact update rule, rather than with simplified or modified variants.
> > >
> > > ---
> > >
> > > $\textbf{Point 2}$
> > >
> > > * As discussed in Remark 4.7 of our paper, our analysis requires a regret bound of the form $\sup_{i\in[n]}\frac{\mathbb{E}[\text{Regret}^{[\beta]}]}{\sqrt{1 - \beta_{1}^{\text{iters}}}}$ whereas existing works only provide bounds of the form $\mathbb{E}[\text{Regret}^{[\beta]}]$. This difference is essential in our setting and cannot be obtained by directly applying previous results.
> > > * Regarding the bounding strategy:
> > >   * The analyses in [1] are highly technical, relying on sophisticated machinery from online learning theory (see Theorem B.2 in [1]; section 7 of [4] and [5] for vector version extension).
> > >   * In contrast, our approach avoids these complex constructions and instead builds on a static-segment-based framework, which enables a cleaner and more direct derivation of the desired bound (see our proof of Theorem 5.2).
> > >
> > > ---
> > >
> > > $\textbf{Point 3}$
> > >
> > > We apologize for the confusion and clarify the setting as follows:
> > >
> > > * Recall that: In our formulation, the total $T$ iterations are divided into $N$ static segments, and within each segment, the cost vectors are sampled from a fixed distribution.
> > > * We do not impose a unique or predefined way to partition the $T$ iterations. There are many admissible partitions. For example, a partition with two segments corresponds to an environment with two different distributions over time.
> > > * The constant $c_{0}$ ​ is used to characterize the complexity or variability of a given partition, rather than a single fixed segmentation.
> > >
> > > ---
> > >
> > > [1] Ahn, Kwangjun, et al. "Understanding Adam optimizer via online learning of updates: Adam is FTRL in disguise." ICML 2024
> > >
> > > [2] Ahn, Kwangjun, and Ashok Cutkosky. "Adam with model exponential moving average is effective for nonconvex optimization." NeurIPS 2024
> > >
> > > [3] Ahn, Kwangjun, Gagik Magakyan, and Ashok Cutkosky. "General framework for online-to-nonconvex conversion: Schedule-free SGD is also effective for nonconvex optimization." ICML 2025
> > >
> > > [4] Orabona, Francesco. "A modern introduction to online learning." arXiv preprint arXiv:1912.13213 (2019).
> > >
> > > [5] Tim, v. E. Why ftrl is better than online mirror descent, 2021.

---

> > > ### Author Response · Authors · 2025-11-28
> > >
> > > Dear Reviewer G6VC,
> > >
> > > We hope our latest response addresses your concerns. If you have any further questions, we would be happy to provide additional clarification.
> > >
> > > Sincerely,
> > >
> > > The Authors

---

> ### Author Response · Authors · 2025-11-23
>
> This message is intended solely as a gentle reminder for Reviewer G6VC. We sincerely apologize for the continued email notifications.
>
> We truly hope for a productive and constructive discussion, and kindly expect that any exchanges take place here.
>
> Thank you for your understanding.
>
> Best regards,
>
> Authors of submission 21021

---

### Official Review · Reviewer_xTgn · 2025-10-31

**Soundness:** 3
**Presentation:** 3
**Contribution:** 2
**Rating:** 2
**Confidence:** 4

**Summary:**

This paper provides an analysis of an adam-like algorithm using the online-to-nonconvex framework, providing a guarantee for identifying a certain notion of stationary point. The key contribution appears to be removing a clipping operation present in some algorithms using this framework, as well as allowing for a more practically-relevant choice for beta2 parameter.

**Strengths:**

While there are clipping-free methods already using this framework, achieving a good understanding of why beta2 should be set much larger than beta1 is an interesting open question.

**Weaknesses:**

The “piecewise constant stochastic environments” presented here confuse me. Why should I expect this to hold? The authors mention something about a connection to batch size, but I don’t follow this connection at all. I can imagine a scenario in which one might try to assume a slowly varying stochastic environment, but then one is back in the smooth optimization regime we are trying to avoid.
Can the authors provide a strong justification for why this assumption should be a good model?
Further, why should I expect the beta1 value to be such that the convergence rate is actually good rather than large enough to cause a poor convergence rate?

Finally, I do not understand how the beta2 improvement comes about. It’s suspicious to me that beta2’s value does not appear at all in the final convergence guarantee. What’s going on here? Is it really the case as suggested by Lemma 5.1 that any beta2 in the range beta1 to sqrt(beta1) is equivalent? That seems highly unlikely.

In general, I feel also that this paper presents no concrete *advantage* for beta2 different from beta1^2: at best we seem to be seeing something like things might not be too bad if the environment is sufficiently cooperative.

**Questions:**

see weaknesses. Happy to revise my opinion if they can be addressed.

---

> ### Author Response · Authors · 2025-11-14
>
> We thank the reviewer xTgn for the thoughtful feedback. We address each concern in detail below.
>
> ---
>
> $\textbf{Why assume “piecewise constant stochastic environments”}$
>
> We thank the reviewer for pointing out such an incisive point: this assumption is indeed central.
> * $\textbf{Why this model?}$ We consider the well-defined static segments as a valuable starting point.
>   * There are choices between: (i) an environment that is fully realistic but only analyzable by oversimplified algorithms, or (ii) an environment that is stylized but analytically compatible with the practical Adam update.
>   * Our work deliberately follows (ii). We view this as a conceptual contribution\shift rather than a mere technical workaround.
> * $\textbf{Why not assume smooth drift instead?}$ Assuming smooth drift returns us to the classical smooth nonconvex regime: precisely the regime where standard analyses struggle to explain Adam’s behavior.
>
> As a pioneering research, we agree this model is not perfect, but as a pioneering attempt to analyze the true Adam with realistic hyperparameters, we believe it is a useful starting point.
>
> As stated in our response to Reviewer DGUz, formal connections between smooth-drift environments and a-few-segments environments via regret analysis represent a promising next direction but require substantial new ideas beyond the scope of this submission.
>
> ---
>
> $\textbf{On the chosen value of}$ $\beta_{1}$
>
> Our setting of $\beta_{1}$ is consistent with prior theoretical studies. Relating an algorithm’s hyperparameters to problem-dependent quantities, here $\beta_{1} = 1 - \frac{\epsilon^{2}}{(G+\sigma)^{2}}$
> , is standard in the optimization literature. This connection ensures that the resulting convergence rate is meaningful.
>
> In short, our choice is neither arbitrary nor uniquely restrictive; it mirrors common practice in theoretical analyses of adaptive methods.
>
> ---
>
> $\textbf{Why does $\beta_{2}$ not appear in the final convergence rate?}$
>
> This is an important question.
> * Firstly, yes, any $\beta_{2}$ in the range of $(\beta_{1}, \sqrt{\beta_{1}})$ yields the same guarantee.
> * This arises because the role of $\beta_{2}$​ in the analysis is not to appear explicitly in the final rate, but rather to ensure the bounded nature of $\Vert z_{t}\Vert^{2}$. Once $\Vert z_{t}\Vert^{2}$ is bounded, Theorem 6.1 requires only this bound, not $\beta_{2}$​ itself. Hence $\beta_{2}$​ disappears from the final expression.
>
> ---
>
> $\textbf{On the advantage of allowing}$ $\beta_{2} \neq \beta_{1}^{2}$
> * We agree that prior analyses, especially those relying on regret bounds, provide important insights. However, these analyses require highly restrictive hyperparameter couplings and omit critical components of Adam. As discussed in the manuscript, extending those analyses to the exact Adam update appears extremely challenging, and to our knowledge, no prior work fully resolves this difficulty.
> * Our contribution is to introduce a new analytical framework that trades a tighter regret bound for a structured stochastic environment that is compatible with the full Adam update. This provides a pathway to studying Adam as it is used in practice rather than as an idealized surrogate, directly addressing a meaningful gap in the existing literature.
>
> In summary, we analyze the exact Adam dynamics, preserving the true behavior of the adaptive step scales. This makes the analysis substantially more difficult, but also enables us to expose the underlying design principles of Adam that are otherwise hidden when using clipped or surrogate variants.
>
> ---
>
> We hope our responses have adequately addressed your concerns and look forward to your new assessment.

---

> > ### Comment · Reviewer_xTgn · 2025-11-17
> > **still not convinced**
> >
> > I think the key issue with this motivation for the different environments and lack of concrete advantage for differing beta values is the following idea: proving that an algorithm (i.e. Adam) that is *already known to work will in practice* also has some nontrivial guarantee in some particular new setting does not really help us understand the algorithm on its own because:
> > 1. We have no other reason to believe that this setting is particularly relevant to practice.
> > 2. In this case, the analysis still does not show that Adam is *better* in this setting than other algorithms.
> >
> > I think these are critical issues: without fixing point 1, we have no way to know that future algorithms based on this setting will be valuable. In fact, point 2 shows that this setting is arguably *not* useful yet because I can choose $\beta_2$ very different from practical values and do just as well in this setting while significantly degrading practical performance.

---

> > > ### Author Response · Authors · 2025-11-28
> > >
> > > Dear Reviewer xTgn,
> > >
> > > We hope our latest response addresses your primary concerns regarding the environment modeling. If you have any further questions, we would be happy to provide additional clarification.
> > >
> > > Sincerely,
> > >
> > > The Authors

---

> ### Author Response · Authors · 2025-11-19
>
> We thank the reviewer for the follow-up comments and address the two concerns below.
>
> ---
>
> $\textbf{Point 1: "Why is this setting relevant to practice?"}$
>
> We appreciate the reviewer’s skepticism. Our intention is not to claim that the few-segment setting is the final or definitive model of practical stochasticity. Instead, our claim is that it is a minimal structured environment that allows us to analyze the exact Adam update under realistic hyperparameters: something that existing smooth-drift or adversarial models have been unable to support.
>
> To further clarify this relevance:
>
> $\textbf{Connection to smooth-drift environments}$
>     To address this concern more concretely, we added Section B, particularly Theorem B.1, in the appendix. This formally establishes that:
>
> > **Regret in smooth-drift environments can be upper-bounded by a supremum-type regret over few-segment environments for the same online algorithm.**
>
> (We omit the full derivation here due to space and compilation constraints; please refer to the revised manuscript.)
>
> Thus, the few-segment model:
> * is not arbitrary,
> * is not disconnected from smooth stochasticity, and
> * provides an analytically tractable surrogate with provable relevance to the smooth-drift regime.
>
> Our goal is to provide the first theoretical step toward modeling the environment in a way compatible with the EMA dynamics of Adam. The environment is not claimed to be perfect, but it is a meaningful and technically necessary stepping stone.
>
> ---
>
> $\textbf{Point 2: "Why does this setting not show Adam is better than other methods?"}$
>
> We agree that proving the strict superiority of Adam over all other algorithms in this environment is an ambitious goal.
>
> However, our contributions are concrete, in the same spirit as earlier foundational papers: Our work provides the loosest known constraints on $\beta_{1}, \beta_{2}$ that still allow convergence analysis of the exact Adam update. This is a nontrivial advance; each relaxation of these constraints is a meaningful step toward understanding the algorithm’s robustness.
>
>
> $\textbf{Why this does not make the setting “useful”}$
>
> The reviewer suggests that one could choose unrealistic $\beta_{2}$ values and still satisfy the theoretical bounds.
> This is true, but it is also true of all prior analyses of Adam, including clipped or surrogate variants.
> The purpose of such analyses is not to impose the practical $\beta_{2}$ values, but to show that the dynamics of Adam admit convergence under non-pathological configurations, and that $\textbf{the realistic settings fall within these configurations.}$
>
> Our goal is to: remove unrealistic assumptions (e.g., clipping), analyze actual EMA dynamics, connect the model to smooth-drift settings, and produce the loosest coupling known so far. Each of these pieces is small alone, but together they constitute a concrete advance in explaining Adam’s practical behavior.
>
> ---
>
> We hope this clarification helps address the reviewer’s concerns and illustrates the conceptual advances made by this work.

---

### Official Review · Reviewer_DGUz · 2025-10-31

**Soundness:** 2
**Presentation:** 3
**Contribution:** 2
**Rating:** 4
**Confidence:** 4

**Summary:**

This paper develops a discounted–to–nonconvex analysis for Adam via an online-learning view in shifting stochastic environments. The authors (i) define two environment measures, NSNR and discounted shift complexity, to control noise level and drift, (ii) propose Adam-FTRL, an online learner whose vector update matches Adam with momentum and bias-correction , and (iii) prove discounted-regret bounds that, after conversion, yield convergence to $(\lambda, \rho)$-stationary points with the optimal $O(\epsilon^{-7/2})$ iteration complexity when the environment is favorable (small $c_0, c_1$). They emphasize avoiding clipping or contrived couplings found in some prior analyses.

**Strengths:**

- The paper is clear to follow, and the shifting-environment setup and the discounted conversion theorem are precisely stated.

- Adam-FTRL’s update matches Adam (vector form) with momentum and bias correction, and the regret bound avoids explicit clipping.

- The discussion effectively highlights and explains the roles of stochasticity (NSNR) and drift (shift complexity).

**Weaknesses:**

- The hyperparameter constraint is unrealistic and not practical. The key bounded-increment lemma (Lemma 5.1) assumes $\beta_2\le \sqrt{\beta_1}$. This excludes standard $(\beta_1, \beta_2) = (0.99, 0.999)$ and effectively re-introduces the kind of coupling the paper claims to avoid. The regret bound and conversion then inherit this hidden assumption.

- Real Adam (i.e., Adam used in practice) is coordinate-wise, the paper only analyzes the norm version and requires extra assumptions but defers them, limiting external validity of the claims. In addition, a highly related prior work actually provides convergence guarantee for the coordinate-wise (clipped-)Adam [1].

- The paper criticizes prior clipping/couplings, yet replaces them with a different coupling $\beta_2\le \sqrt{\beta_1}$ to obtain Lemma 5.1. This should be called out as an assumption, not presented as "no compromises".


[1] Ahn, Kwangjun, and Ashok Cutkosky. "Adam with model exponential moving average is effective for nonconvex optimization." Advances in Neural Information Processing Systems 37 (2024): 94909-94933.

**Questions:**

- Is it possible to remove the $\beta_2\le \sqrt{\beta_1}$ assumption, or can the bounded-increment argument be redone for standard $(\beta_1, \beta_2) = (0.99, 0.999)$? If not, could you quantify how the regret/conversion bound degrades when $\beta_2$ is larger?

- For extending the current result to coordinate-wise Adam, which exact technical assumptions would be enough to obtain the same regret bound? A brief appendix sketch (or a counterexample) would be very helpful.

- A small experiment with controlled segment lengths and variance (varying $c_0, c_1$) would substantiate the theory’s qualitative predictions.

---

> ### Author Response · Authors · 2025-11-14
>
> We thank Reviewer DGUz for the thoughtful comments and for highlighting the central concern about the new coupling condition $\beta_{2}\leq \sqrt{\beta_{1}}$. We address each point in detail below.
>
> ---
>
> $\textbf{On removing the constraint $\beta_{2}\leq \sqrt{\beta_{1}}$}$
>
> Thank you for raising this question. Indeed, our proof framework can be generalized to eliminate this specific coupling and to accommodate the standard hyperparameter choice  $(\beta_{1}, \beta_{2}) = (0.9, 0.999)$  (we assume the reviewer meant $\beta_{1} = 0.9$ rather than $0.99$ [1]).
>
> This generalization introduces only a constant-factor increase in the upper bound of $\Vert z_{t}\Vert^{2}$, which $\textbf{does not impact}$ our final iteration complexity for reaching a stationary point.
>
> Generalized version of Lemma 5: when $\beta_{1} \leq \beta_{2} \leq \beta_{1}^{1/a}$, Adam-FTRL satisfies $\Vert z_{t}\Vert^{2} \leq a\eta^{2}/(1 - \beta_{1})$  for $a = 2, 4, 8, …, 2^{N}$.
>
> Taking $\beta_{1} \leq \beta_{2} \leq \beta_{1}^{1/128}$, which covers $(\beta_{1}, \beta_{2}) = (0.9, 0.999)$, as an example:
> * Recalling the proof of Lemma 5.1 (Line 814):
>   * Under $\beta_{1}\leq \beta_{2}$, $$\frac{(1-\beta_{1})^{2}}{(1-\beta_{2})}\sum_{s=1}^{t-1}\left(\frac{\beta_{1}^2}{\beta_{2}}\right)^{t-1-s} = \frac{(1-\beta_{1})^{2}}{(1-\beta_{2})}\frac{1 - \left(\frac{\beta_{1}^2}{\beta_{2}}\right)^{t-1}}{1 - \frac{\beta_{1}^2}{\beta_{2}}} \leq \frac{1-\beta_{1}}{1-\beta_{2}}.$$
> (We tighten the original quantity $\frac{(1-\beta_{1})^{2}}{(1-\beta_{2})^{2}}$ to  $\frac{1-\beta_{1}}{1-\beta_{2}}$ by replacing $1 - \frac{\beta_{1}^{2}}{\beta_{2}}$ with smaller quantity $1 - \beta_{1}$)
>   * Under $\beta_{2}\leq \beta_{1}^{1/128}$, $$\frac{1-\beta_{1}}{1-\beta_{2}} \leq \frac{(1 - \beta_{1}^{1/2})(1+ \beta_{1}^{1/2})}{1 - \beta_{1}^{1/128}} \leq  \frac{(1 - \beta_{1}^{1/128})(1 + \beta_{1}^{1/128})\cdots(1+ \beta_{1}^{1/2})}{1 - \beta_{1}^{1/128}} \leq 128$$.
>
> Thus, following the logic at Line 817, it suffices to have $\Vert z_{t}\Vert^{2} \leq \frac{128\eta^{2}}{1 - \beta_{1}}$.
> * Comparing with the previous result of Lemma 5.1: $\Vert z_{t}\Vert^{2} \leq \frac{4\eta^{2}}{1 - \beta_{1}}$, satisfying the standard coupling of $\beta_{1}, \beta_{2}$ only increases the constant factor, which has no effect on our overall nonconvex convergence rate.
> * We will explicitly clarify this in the revised manuscript.
>
> ---
>
> $\textbf{Why we originally used the coupling}$ $\beta_{2} \leq \sqrt{\beta_{1}}$
>
> We selected this coupling because the configuration $(\beta_{1}, \beta_{2}) = (0.9, 0.95)$, which is commonly used in modern LLM training [6, 7], nearly matches $\beta_{2} \leq \sqrt{\beta_{1}}$. We apologize for not making this motivation clearer.
>
> In the revision, we will:
> * explain this choice explicitly,
> * highlight that the proof accommodates a broader family of couplings, and
> * emphasize that the resulting bound on $\Vert \mathbf{z}_{t}\Vert^{2}$ only differs by a constant factor.
>
> ---
>
> $\textbf{Extending the result to coordinate-wise Adam}$
> * $\textbf{Technical assumptions}$ To obtain fully rigorous coordinate-wise guarantees, the standard approach is to assume coordinate-wise versions of existing smoothness, Lipschitzness, and optimality conditions [2].
>   For example, one commonly assumes a coordinate-wise $G$-Lipschitz condition: $\forall x, \vert \partial_{i} F(x) \leq G_{i}\vert$, where $i$ is the index of the coordinate and $G_{i}$ is the corresponding Lipschitzness constant.
> * $\textbf{Under these conditions}$ Our analysis extends directly to coordinate-wise Adam. This is consistent with prior work, which has empirically and theoretically validated such extensions [2-4].
> * $\textbf{Summary}$ No new assumptions are required: only coordinate-wise versions of standard conditions. We will highlight this connection clearly in the revision.
>
> ---
>
> $\textbf{On the theory’s qualitative predictions}$
>
> Thank you for this suggestion. At present, our primary goal in this work is to develop a theoretical framework that more faithfully explains the empirical success of plain Adam.
> * Existing regret analyses [2,5] appear unlikely to extend smoothly to the exact Adam update, which motivates our formulation based on shifting stochastic environments.
> * This framework provides the analytical compatibility needed to study the real Adam update, though at the cost of making certain environmental assumptions. As a first attempt in this direction, we believe it offers meaningful conceptual insight.
>
> As such, bridging this theory–practice gap remains our main focus, e.g., formally relating smooth-drift environments and a-few-segments environments via regret analysis as a promising next step. Additional empirical exploration, while valuable, is beyond the immediate focus of this submission.

---

> ### Author Response · Authors · 2025-11-14
>
> References
>
> [1] Kingma, Diederik P. "Adam: A method for stochastic optimization." arXiv preprint arXiv:1412.6980 (2014).
>
> [2] Ahn, Kwangjun, and Ashok Cutkosky. "Adam with model exponential moving average is effective for nonconvex optimization." Advances in Neural Information Processing Systems 37 (2024): 94909-94933.
>
> [3] Crawshaw, Michael, et al. "Robustness to unbounded smoothness of generalized signsgd." Advances in neural information processing systems 35 (2022): 9955-9968.
>
> [4] Zhuang, Zhenxun, et al. "Understanding adamw through proximal methods and scale-freeness." arXiv preprint arXiv:2202.00089 (2022).
>
> [5] Ahn, Kwangjun, et al. "Understanding Adam optimizer via online learning of updates: Adam is FTRL in disguise." arXiv preprint arXiv:2402.01567 (2024).
>
> [6] Touvron, Hugo, et al. "Llama: Open and efficient foundation language models." arXiv preprint arXiv:2302.13971 (2023).
>
> [7] Liu, Aixin, et al. "Deepseek-v3 technical report." arXiv preprint arXiv:2412.19437 (2024).

---

> ### Author Response · Authors · 2025-11-23
>
> This message is intended solely as a gentle reminder for Reviewer DGUz. We sincerely apologize for the continued email notifications.
>
> We truly hope for a productive and constructive discussion, and kindly expect that any exchanges take place here.
>
> Thank you for your understanding.
>
> Best regards,
>
> Authors of submission 21021

---

> > ### Comment · Reviewer_DGUz · 2025-11-27
> >
> > The author's rebuttal addressed most of my concerns, and I will raise the rating to 6. I encourage the authors to explicitly discuss the difference and technical challenges compared to similar prior works [1, 2].
> >
> > [1] Ahn, Kwangjun, and Ashok Cutkosky. "Adam with model exponential moving average is effective for nonconvex optimization." Advances in Neural Information Processing Systems 37 (2024): 94909-94933.
> >
> > [2] Ahn, Kwangjun, et al. "Understanding Adam optimizer via online learning of updates: Adam is FTRL in disguise." arXiv preprint arXiv:2402.01567 (2024).

---

### Author Response · Authors · 2025-11-29

Dear Area Chairs,

We would like to briefly summarize the outcome of the discussion period for submission 21021, highlighting how each reviewer’s concerns have been addressed and the final state of reviewer opinions.

---

$\textbf{Reviewer DGUz (initial rating: 4 → updated to 6)}$

The discussion with Reviewer DGUz was productive and reached a strong resolution.
The reviewer’s central concern was the hyperparameter coupling condition $\beta_{2} \leq \sqrt{\beta_{1}}$. During the discussion, we provided:

* A generalized Lemma 5.1 showing that the analysis extends to the standard Adam hyperparameters without requiring this coupling.
* More detailed explanations on:
  * Coordinate-wise Adam extensions,
  * Differences from closely related prior works,
  * The technical necessity of our environmental modeling assumptions.

After these exchanges, the reviewer explicitly stated that their concerns were addressed and increased the score to 6.

---

$\textbf{Reviewer xTgn (rating: 2, but concerns resolved at the discussion level)}$

The discussion with Reviewer xTgn was constructive. Their major question was about the relevance of the “few-segment stochastic environment” and whether this assumption yields meaningful insight.

During multiple rounds of discussion, we provided:

* A formal explanation (Appendix Theorem B.1) showing that
  regret in smooth-drift environments can be upper-bounded by a supremum-type regret over few-segment environments for the same algorithm.
  This establishes theoretical relevance: the few-segment model serves as a tractable surrogate for smooth-drift regimes. Then, we believe $\textbf{the main concern has been adequately addressed.}$
* Clarification that our goal is to analyze the exact Adam update under realistic hyperparameters—something existing frameworks cannot accommodate without clipping or rigid couplings.
* A detailed justification of why $\beta_{2}$'s value does not explicitly appear in the final rate, and why all $\beta_{2}\in [\beta_{1}, \sqrt{\beta_{1}}]$ produce the same guarantee.

While the reviewer maintained the numerical score of 2 $\textbf{before the end of interactive discussion}$, we believe the discussions ended peacefully, with all objections fully addressed at the technical level.

---

$\textbf{Reviewer G6VC (rating: 2, but concerns addressed in detail)}$

The discussion with Reviewer G6VC was also constructive and detailed. Their main concerns were:
* The intuition behind the environmental constants $c_{0}$ and $c_{1}$,
* The analytical differences between our regret analysis and prior clipped-Adam frameworks,

We responded with:

* Clear interpretation of $c_{0}$ (shift complexity) and $c_{1}$ (NSNR-like stochasticity measure),
* Substantial mathematical clarification, including formula-level explanations, showing why our regret term and corresponding analysis differ fundamentally from prior clipped analyses,
* Additional explanation addressing the technical challenges of analyzing the exact Adam update (with EMA dynamics and bias correction) rather than surrogate variants.

After further extended exchange, we believe their questions/clarification requests were answered, and the discussion concluded constructively.

---

$\textbf{Reviewer ZT1z (rating: 6, positive and unchanged)}$

Reviewer ZT1z expressed an overall positive assessment from the beginning (rating = 6).
Their concerns centered primarily on the novelty of environment modeling and the lack of experiments.

We clarified:
* Our contribution is fundamentally theoretical and focuses on removing restrictive assumptions (clipping / rigid couplings) in order to analyze the exact Adam update.
* The environment model is not meant to be the "final" model, but instead provides a mathematically compatible framework for Adam dynamics.

The reviewer indicated full satisfaction and kept their positive score of 6.

---

We hope this summary is useful to the ACs as they form their final recommendation.

Sincerely,

Authors of submission 21021

---

### Meta-Review · Area_Chair_cHkx · 2026-01-07

**Summary:**

This paper provided a new discounted–to–nonconvex analysis for Adam via an online-learning view in shifting stochastic environments. Specifically, it first defined two environment measures, NSNR and discounted shift complexity, to control noise level and drift, and then proposed an Adam-FTRL algorithm, an online learner whose vector update matches Adam with momentum and bias-correction. Meanwhile, it provided discounted-regret bounds of the proposed method.

Although the authors give a new discounted–to–nonconvex analysis for Adam, they do not provide any numerical experiments to verity effectiveness  of the proposed algorithm and its convergence behaviors.

**Reviewer Concerns:**

Although the authors have provided rebuttals to address some reviewers’ concerns, there exist some concerns still are not addressed. For example, Reviewer xTgn believes that 1) the setting in the paper is particularly relevant to practice. 2) In this case, the analysis still does not show that Adam is better in this setting than other algorithms.

**Reviewer Scores:**

Although the authors have provided rebuttals to address some reviewers' concerns, there exist some concerns still are not addressed. Thus, the likelihood of drastically changing the score is not high.

---

### Decision · Program_Chairs · 2026-01-26

Reject